  

# Cyclin A2 localises in the cytoplasm at the S/G2 transition to activate PLK1

Helena Silva Cascales[1] ⓘ, Kamila Burdova[2], Anna Middleton[1] ⓘ, Vladislav Kuzin[1] ⓘ, Erik Müllers[1] ⓘ, Henriette Stoy[1], Laura Baranello[1] ⓘ, Libor Macurek[2] ⓘ, Arne Lindqvist[1] ⓘ

**Cyclin A2 is a key regulator of the cell cycle, implicated both in DNA replication and mitotic entry. Cyclin A2 participates in feedback loops that activate mitotic kinases in G2 phase, but why active Cyclin A2-CDK2 during the S phase does not trigger mitotic kinase activation remains unclear. Here, we describe a change in localisation of Cyclin A2 from being only nuclear to both nuclear and cytoplasmic at the S/G2 border. We find that Cyclin A2-CDK2 can activate the mitotic kinase PLK1 through phosphorylation of Bora, and that only cytoplasmic Cyclin A2 interacts with Bora and PLK1. Expression of predominately cytoplasmic Cyclin A2 or phospho-mimicking PLK1 T210D can partially rescue a G2 arrest caused by Cyclin A2 depletion. Cytoplasmic presence of Cyclin A2 is restricted by p21, in particular after DNA damage. Cyclin A2 chromatin association during DNA replication and additional mechanisms contribute to Cyclin A2 localisation change in the G2 phase. We find no evidence that such mechanisms involve G2 feedback loops and suggest that cytoplasmic appearance of Cyclin A2 at the S/G2 transition functions as a trigger for mitotic kinase activation.**

## Introduction

Correct progression through the cell cycle depends on the tight regulation of Cyclin–Cyclin dependent kinase (CDK) complexes over time. Sequential waves of CDK activity ensure timely phosphorylation of a large number of substrates. CDK activity increases through the cell cycle, allowing separation of high-affinity sites that initiate early events as onset of DNA replication, and low-affinity sites that enable late events as mitosis (Swaffer et al, 2016). On top of this general trend, different Cyclin–CDK complexes can selectively phosphorylate target proteins, both by different affinity to a target and by being present at different cellular locations (Örd & Loog, 2019). For example, although both Cyclin A2 (CycA2) and Cyclin B1 (CycB1) have been implicated in mitotic entry, CycA2 appears mostly nuclear, whereas CycB1 appears mainly in the cytoplasm (Pines & Hunter, 1991).

Because of its presence during S, G2, and early mitosis, CycA2 is at a strategic position to control a large part of the cell cycle (Pagano et al, 1992; Fung et al, 2007). Indeed, CycA2 regulates multiple aspects of the S phase by steering phosphorylation of key components as CDC6 (Petersen et al, 1999), pre-replication complexes (Furuno et al, 1999; Katsuno et al, 2009), and components of the replication machinery (Cardoso et al, 1993; Frouin et al, 2005). Depletion of CycA2 leads to an arrest in G2 phase, which could be suspected because of errors in completing DNA replication (Fung et al, 2007; Gong et al, 2007; De Boer et al, 2008; Gong & Ferrell, 2010; Oakes et al, 2014). However, recent data show that depletion of CycA2 specifically in G2 phase blocks mitotic entry. Interestingly, similar depletion of CycB1 does not block mitotic entry, but rather affects later mitotic progression (Hégarat et al, 2020). This shows that apart from its functions during S phase, CycA2 is a key player in regulating progression into mitosis. Similar to many cyclins, however, mouse embryonic fibroblasts lacking CycA2 can be isolated, suggesting that its absence can be compensated by other Cyclins (Kalaszczynska et al, 2009; Satyanarayana & Kaldis, 2009).

During G2 phase, CycA2 stimulates transcription and represses degradation of multiple mitotic regulators (Lukas et al, 1999; Laoukili et al, 2008; Oakes et al, 2014; Hein & Nilsson, 2016). As the mitotic regulators accumulate, CycA2 participates in the feedback loops that culminate in full CDK1 activation and mitotic entry (Mitra & Enders, 2004). A key player in these feedback loops is Polo-like kinase 1 (PLK1) (Lindqvist et al, 2009). We and others previously showed that PLK1 is activated by Aurora A, in a reaction that requires the cofactor Bora (Macurek et al, 2008; Seki et al, 2008). CDK-mediated phosphorylation of Bora facilitates PLK1 activation, and both CycA2- and CycB1-containing complexes have been suggested to phosphorylate Bora (Parrilla et al, 2016; Thomas et al, 2016; Gheghiani et al, 2017; Vigneron et al, 2018). Interestingly, Bora appears exclusively cytoplasmic, raising the possibility that a cytoplasmic Cyclin–CDK activates PLK1 (Feine et al, 2014; Bruinsma et al, 2015).

Although CycA2 targets are mostly nuclear, CycA2 has been shown to also regulate events in the cytoplasm, particularly after the S phase. This includes loading of Eg5 to centrosomes in the G2 phase and inhibiting endocytic vesicle fusion to control membrane transport as cells enter into mitosis (Woodman et al, 1993; Kanakkanthara

[1]Department of Cell and Molecular Biology, Karolinska Institutet, Stockholm, Sweden  [2]Laboratory of Cancer Cell Biology, Institute of Molecular Genetics, Academy of Sciences of the Czech Republic, Prague, Czech Republic

Correspondence: arne.lindqvist@ki.se

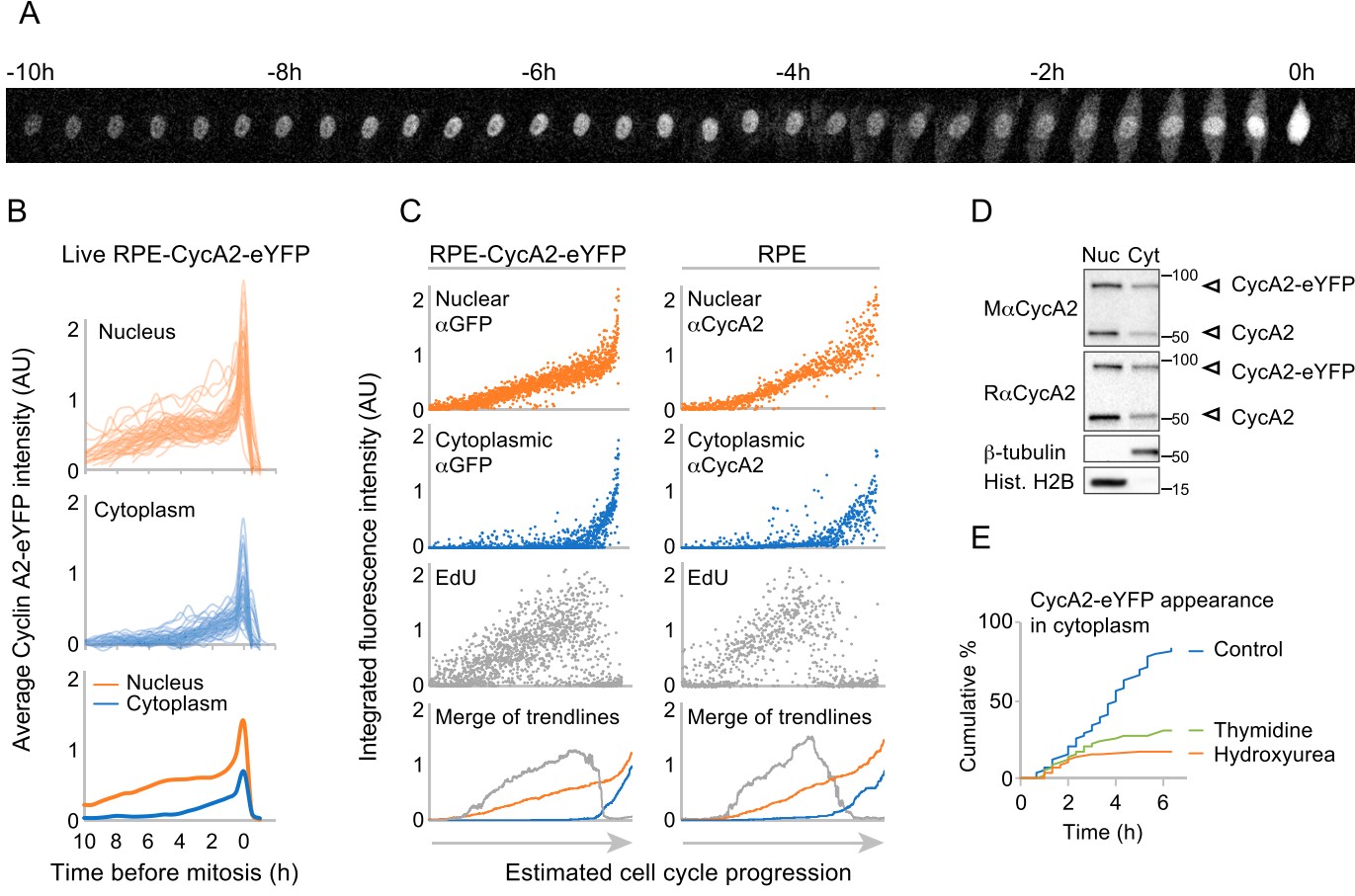

**Figure 1. Cyclin A2 accumulates in the cytoplasm at the S/G2 transition.**
**(A)** Time-lapse imaging through mitosis of a single RPE cell gene-targeted to express CycA2-eYFP. Time between images is 20 min. **(B)** Quantification of mean fluorescence intensity in the nucleus (top) and in the cytoplasm (middle) of 50 individual RPE CycA2-eYFP cells over time. Cells were synchronised in silico to set t = 0 at mitosis. Nuclear and cytoplasmic fluorescence in mitosis is approximated to central and peripheral regions of the mitotic cell to show a continuous graph. Please note that the increase in mean intensity during mitosis largely is due to cell rounding. Bottom graph shows the average of nuclear and cytoplasmic mean intensities. **(C)** Time-course estimate from fixed cells, pulse-labelled with EdU. 2,100 RPE-CycA2-eYFP cells were sorted based on increasing anti-GFP and DAPI staining (left) and 1,567 RPE cells were sorted based on increasing anti-CycA2 and DAPI staining (right). Graphs focus on area where CycA2 expression is apparent. Lower graphs show running median of 100 (left) or 50 (right) cells. **(D)** Western blot of nuclear and cytoplasmic fractions of unsynchronised RPE-CycA2-eYFP cells using the indicated antibodies. **(E)** Quantification of the percentage of RPE-CycA2-eYFP cells accumulating CycA2-eYFP in the cytoplasm after different treatments. Cells were treated with DMSO (control), Thymidine or Hydroxyurea and immediately imaged. The number of cells accumulating CycA2 in the cytoplasm was recorded and plotted as a percentage of the total number of cells tracked. All experiments were repeated at least three times.

et al, 2016; Li et al, 2018). CycA2 appears predominantly nuclear but can shuttle between nucleus and cytoplasm, despite not possessing a classical NLS (Jackman et al, 2002). A clear cytoplasmic presence of CycA2 is only visible during the G2 phase (Zerjatke et al, 2017). CycA2 association with different proteins has been described to affect localisation both to the nucleus and to the cytoplasm (Maridor et al, 1993; Tsang et al, 2007; Bendris et al, 2011). However, the exact mechanism that regulates CycA2 localisation remains elusive. In addition, whether CycA2 localisation impacts on cell cycle progression remains unclear.

Here, we characterise CycA2 localisation through the cell cycle and describe that CycA2 appears in the cytoplasm at the S/G2 transition. We find that cytoplasmic CycA2 activates PLK1 through phosphorylation of Bora. We further describe that cytoplasmic localisation of CycA2 is abolished in response to DNA damage in a manner that depends on p21. The cytoplasmic appearance of CycA2 depends on multiple mechanisms, including CycA2 association with chromatin during DNA replication. We find no evidence that these

mechanisms involve feedback loops involved in PLK1 and CDK1 activation, raising the possibility that cytoplasmic localisation of CycA2 functions as a trigger for mitotic kinase activation.

## Results

### Cyclin A2 accumulates in the cytoplasm at the S/G2 transition

To study the dynamics of CycA2 in live cells we targeted *CCNA2* using rAAV-mediated homologous recombination. We introduced an ORF for enhanced (E)YFP in the *CCNA2* locus of U2OS (Akopyan et al, 2014) and retinal pigment epithelium (RPE) cell lines to create a CycA2-eYFP fusion protein (Figs 1A and S1A and B). Western blot analysis confirmed the successful integration of the eYFP ORF in one of the two alleles of CycA2 as we detected a band that migrated at the predicted size of endogenous untagged CycA2 and a band

that migrated at the predicted size of the CycA2-eYFP fusion protein (Fig S1A). Importantly, siRNA to target CycA2, addition of S-trityl-L-cysteine, or addition of etoposide showed a similar behaviour of both bands as to CycA2 in parental RPE cells, indicating that eYFP was specifically introduced at the *CCNA2* locus (Fig S1A). Similar to in U2OS cells, CycA2-eYFP in RPE cells is present in all cells from early S phase and its levels increase over time reaching a maximum at mitosis when CycA2 is rapidly degraded (Figs 1A and S1B) (Akopyan et al, 2014).

To monitor CycA2-eYFP localisation, we followed RPE CycA2-eYFP and U2OS CycA2-eYFP cells by time-lapse microscopy (Figs 1A and S1B). In all cells, CycA2-eYFP was first visible in the nucleus, but also appeared in the cytoplasm ~4 h before mitosis (Fig 1B; please note that the apparent fluorescence increase in mitosis is due to cell rounding). We next treated RPE CycA2-eYFP cells with a short pulse of EdU to mark cells in the S phase and quantified immunofluorescence stainings. Whereas CycA2-eYFP was present in the nucleus in all cells once expressed, cells positive for cytoplasmic CycA2-eYFP contained 4n DNA content and low EdU staining (Fig S1C). A similar pattern was found when labelling the parental RPE cells, using an antibody to detect endogenous CycA2 (Fig S1D). We note that CycA2-eYFP showed a slightly higher expression in the cytoplasmic fraction compared with CycA2. However, although the magnitude of cytoplasmic accumulation may differ, the pattern and timing of cytoplasmic appearance is similar for CycA2 and CycA2-eYFP, showing that CycA2-eYFP can be used to study CycA2 localisation (Figs 1C and D and S1C and D).

To pinpoint the position in the cell cycle when cells start to accumulate cytoplasmic CycA2, we estimated a time-course by sorting cells for increasing CycA2 or CycA2-eYFP levels and DAPI content (Akopyan et al, 2014, 2016). Interestingly, we found that CycA2-eYFP appeared in the cytoplasm at the time when a decrease in EdU incorporation was apparent (Fig 1C). A similar pattern was found when assessing CycA2 levels in parental RPE cells (Fig 1C). If CycA2 appears in the cytoplasm after completion of S phase, we reasoned that blocking progression through the S phase would inhibit cytoplasmic appearance of CycA2. We therefore treated RPE-CycA2-eYFP cells with thymidine or hydroxyurea and quantified the proportion of cells that accumulate CycA2-eYFP in the cytoplasm (Fig 1E). The treatment with either drug resulted in a decreased number of cells accumulating CycA2-eYFP in the cytoplasm, suggesting that cells blocked in the S phase do not gain cytoplasmic CycA2-eYFP. Thus, CycA2 appears in the cytoplasm at the S/G2 transition and gradually increases in the cytoplasm through the G2 phase.

## CycA2 regulates the S/G2 transition

We next sought to test whether cytoplasmic appearance of CycA2 impacts on cell cycle progression. CycA2 has functions through both the S and G2 phases and in accordance with previous reports, we find that knockdown of CycA2 leads to accumulation of 4N cells.

To separate events in the S phase and G2, we depleted CycA2 by siRNA and monitored S phase progression by a proliferating cell nuclear antigen (PCNA) chromobody and PLK1 activation by a Förster resonance energy transfer (FRET)-based biosensor (Akopyan et al, 2014). Whereas control cells showed PLK1 activation as PCNA foci sharply decreased at the S/G2 border, CycA2-depleted cells showed no sharp decrease in PCNA foci. Rather, the amount and

intensity of PCNA foci gradually decreased, and PLK1 activity remained low (Fig 2A and B). This shows that PLK1 activation is impaired after CycA2 depletion, either directly or indirectly as the S/G2 transition is impaired in the absence of CycA2. To test whether cytoplasmic CycA2 can rescue cell cycle progression after depletion of endogenous CycA2, we constructed siRNA-resistant versions of CycA2 (CycA2-NT) tagged to mCherry, including either three NES sequences or six NLS sequences (Fig 2C). We used inducible expression of the CycA2 constructs in p53$^{-/-}$ RPE cells to avoid possible nuclear relocation due to p53-p21 activation after CycA2 siRNA (see below). Although both CycA2-mCherry 3×NES and 6×NLS are not restricted from nucleo-cytoplasmic shuttling, we find that upon induction, CycA2-mCherry 3×NES appears predominately in the cytoplasm and CycA2-mCherry 6×NLS appears predominately in the nucleus (Fig 2D). We find that both constructs partially rescued a cell cycle delay after CycA2 siRNA, suggesting that the CycA2 siRNA phenotype depends on a reduction of CycA2 levels (Figs 2E and S2). Interestingly, expression of CycA2-mCherry 3xNES stimulated mitotic entry earlier than CycA2-mCherry 6×NLS, which would be consistent with a function for cytoplasmic CycA2 in stimulating cell cycle progression.

## Cytoplasmic CycA2 triggers PLK1 activation

CycA2 is recently suggested to promote mitotic entry by activation of PLK1 (Gheghiani et al, 2017; Vigneron et al, 2018). As depletion of CycA2 resulted in a loss of PLK1 activation, but also suggested that S phase completion may be impaired (Fig 2A and B), we sought to investigate the contribution of Cyclin–CDK complexes after the S/G2 transition. To this end, we added inhibitors to CDK1 or CDK2 to cells in the G2 phase and monitored PLK1 activation. We find that addition of either CDK1 or CDK2 inhibitor disturbed PLK1 activity as well as the pT210 modification of PLK1, showing the most prominent effect with a combination of both inhibitors (Fig 3A and B). CDK1 and CDK2 are structurally related, and we cannot exclude the possibility that the inhibitor of one CDK in our setup to some extent affects the other CDK. Furthermore, the data do not exclude that the inhibitors target other kinases in addition to CDKs. Nonetheless, a possible explanation of our results is that CDK activity is important for PLK1 activation during G2 phase. Indeed, Aurora A-mediated phosphorylation of PLK1 was further stimulated by CycA2-CDK2 activity in the presence of Bora (Fig 3C). Similarly, we find that CycA2-CDK2 was able to phosphorylate Bora at a Thr–Pro motif and possibly other residues in vitro (Fig 3D). CDK1-mediated phosphorylation of Bora has been characterized previously, including phosphorylation of multiple Thr–Pro motifs, and considering the similarities between CycA2-CDK1 and CycA2-CDK2 we consider it likely that similar residues are phosphorylated (Feine et al, 2014; Tavernier et al, 2015; Thomas et al, 2016; Vigneron et al, 2018). To assess if and where CycA2 forms a complex with PLK1 and Bora, we immunoprecipitated CycA2 from nuclear or cytoplasmic extracts of G2 cells and probed for interactors. Interestingly, we found that both Bora and PLK1 co-immunoprecipitated with CycA2 specifically in the cytoplasmic fraction of G2 synchronised cells (Fig 3E). Similarly, an interaction between Bora and PLK1 was detected exclusively in the cytoplasm (Fig 3F). Thus, CycA2-CDK2 can stimulate phosphorylation of PLK1 T210 in vitro and the interactions required for CycA2-mediated PLK1 T210 phosphorylation are detected in the cytoplasm, but not in the nucleus.

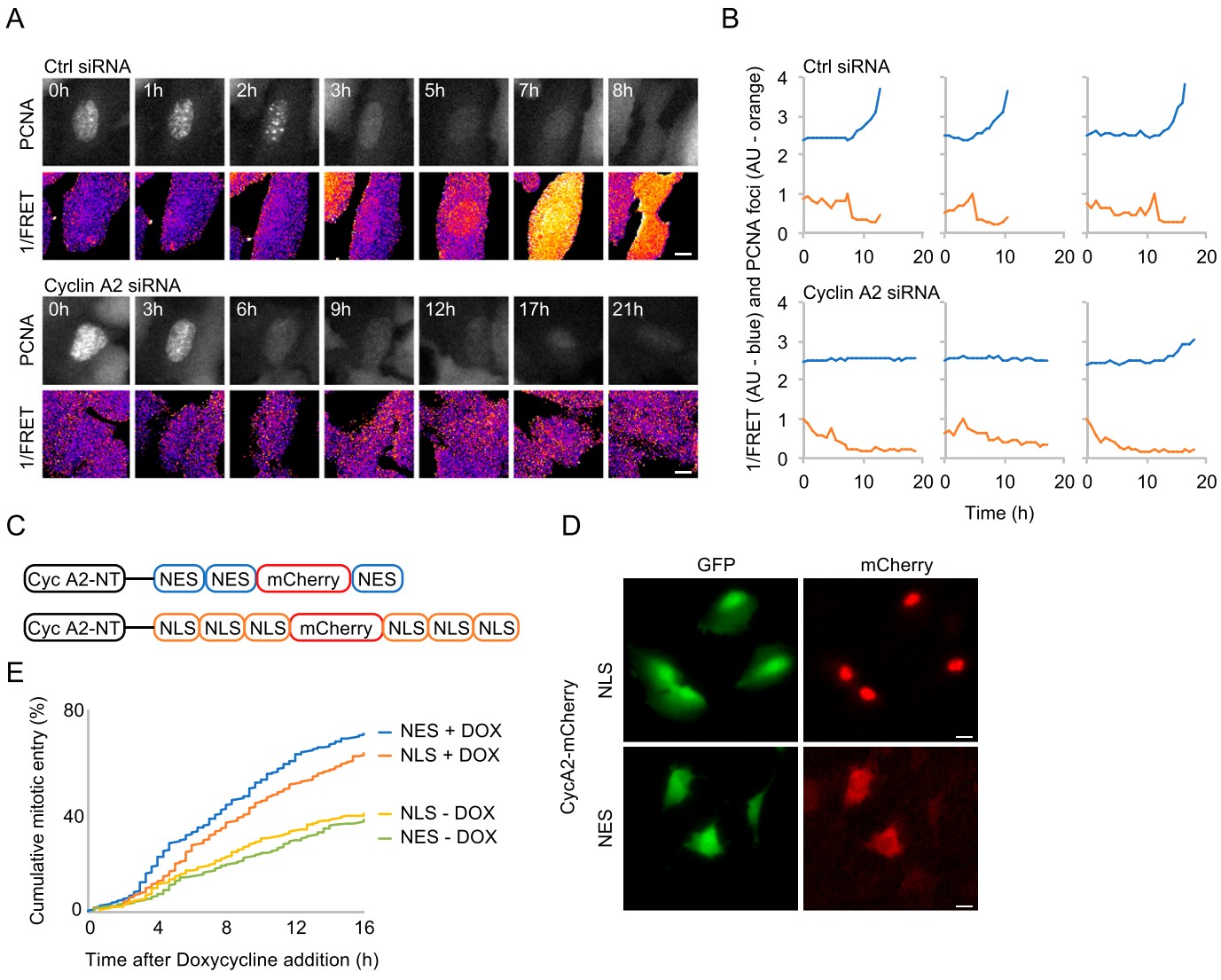

**Figure 2. CycA2 containing nuclear export sequences can rescue cell cycle arrest after depletion of endogenous CycA2.**
**(A)** Time-lapse sequence of U2OS cells expressing polo-like kinase 1 FRET reporter and PCNA chromobody. Time points (h) are indicated in figure. Top Ctrl siRNA, bottom CycA2 siRNA. Scale bar 10 $\mu$m. **(B)** Representative quantifications of individual cells, imaged as in (A). Red line shows 1/FRET and blue line shows PCNA foci. Please note that both pattern of polo-like kinase 1 activation and pattern of PCNA foci are altered after CycA2 siRNA. PCNA foci fluorescence is normalised on the maximal value, and 1/FRET is normalised on the minimal value. **(C)** Schematic of siRNA-non-targetable CycA2 expression constructs. **(D)** Images show RPE p53$^{-/-}$ TetON OsTIR, co-transfected with Piggy-BAC transposase, GFP flanked by Piggy-BAC integration sites (to mark transfected cells) and the constructs outlined in (C) after 18 h Doxycycline addition. Scale bar 20 $\mu$m. **(E)** Quantification of mitotic entry in GFP-expressing cells. Unsynchronised polyclonal RPE p53$^{-/-}$ TetON OsTIR populations treated as in (D) were transfected with CycA2 siRNA. 48 h later, cells were transferred to L15 medium with or without Doxycycline and GFP-expressing cells were followed by time-lapse microscopy. At least 275 cells were followed per condition.

To test the importance of CycA2-mediated PLK1 activation for cell cycle progression, we monitored mitotic entry in a U2OS cell line with inducible expression of PLK1 containing the phospho-mimicking substitution T210D. Induction of PLK1 T210D did not affect the rate of mitotic entry, suggesting that phosphorylation of T210 is not a rate-limiting factor for mitotic entry. After transfection of siRNA to CycA2, mitotic entry was impaired. Importantly, we found that induced expression of PLK1 T210D partially rescued mitotic entry after CycA2 siRNA, suggesting that CycA2 stimulates mitotic entry in part by activating PLK1 (Fig 3G). Thus, although CycA-CDK is active through the S phase, PLK1 activation is detected simultaneously as CycA2 appears in the cytoplasm at the S/G2 transition. Depletion of CycA2 leads to an arrest in G2 phase with low

PLK1 activity, but mitotic entry can be partially rescued by expression of largely cytoplasmic CycA2 or by active PLK1. Inhibitors of CDK activity reduce PLK1 activation in G2 phase and interactions between CycA2, Bora, and PLK1 are detected in the cytoplasm but not in the nucleus. Taken together, our results indicate that cytoplasmic CycA2 participates in the initial activation of PLK1 during G2.

## Cytoplasmic accumulation of CycA2 is not triggered by mitotic kinases

The observation that CycA2-eYFP only accumulates in the cytoplasm after S phase completion suggests that cytoplasmic localisation is

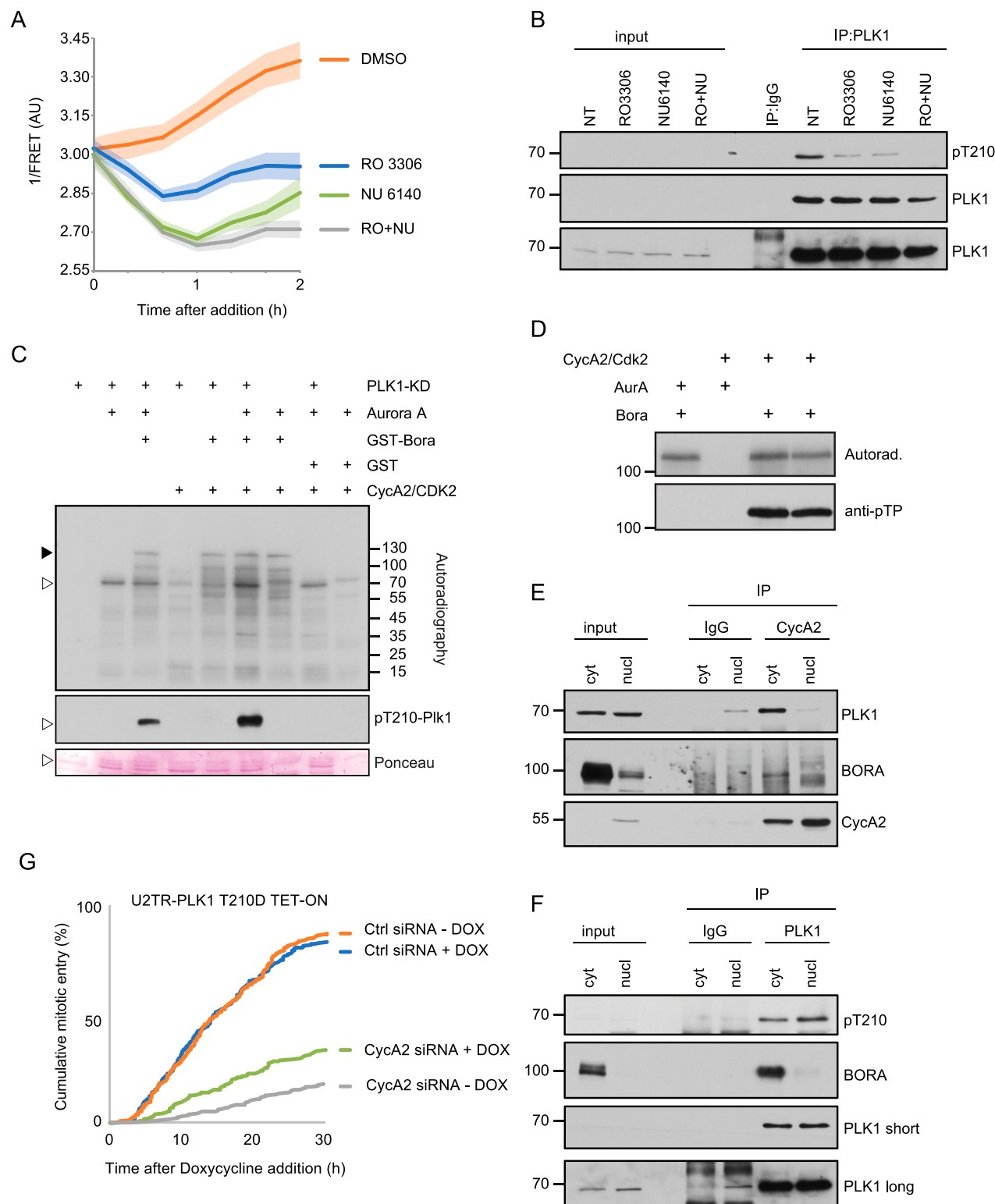

**Figure 3. Cytoplasmic CycA2 triggers polo-like kinase 1 (PLK1) activation.**
**(A)** Inhibition of cyclin dependent kinase (CDK) activity in U2OS cells expressing PLK1 FRET reporter. Cells with intermediate PLK1 FRET signal, indicative of G2 phase, were followed after addition of indicated inhibitors. Graph shows average and s.e.m of at least 10 cells per condition. **(B)** Inhibition of CDK activity in RPE cells synchronised in G2 decreases the level of PLK1 phosphorylation at T210. PLK1 blot is shown twice using a high and a low exposure. **(C)** Phosphorylation of Bora by CycA2-CDK2 promotes modification of PLK1 at T210 mediated by Aurora-A. Empty arrowhead indicates position of the kinase dead PLK1, full arrowhead indicates position of Bora. **(D)** CycA2-CDK2 can phosphorylate Bora on SP/TP sites. **(E, F)** Cytosolic and nuclear extracts from RPE cells synchronised in G2 phase were subjected to immunoprecipitation with

suppressed during the S phase or stimulated during the G2 phase. We previously showed that mitotic-inducing activities of CDK1 and PLK1 start to accumulate at the S/G2 transition (Akopyan et al, 2014). However, we find no evidence that addition of inhibitors to PLK1 or its upstream kinase Aurora A affects cytoplasmic appearance of CycA2 (not shown). Addition of inhibitors to CDK1 or CDK2 led to a slight decrease in cells accumulating cytoplasmic CycA2-eYFP, but interpretation of these results is hampered by that CDK inhibition may affect the S phase progression (not shown). To assess the contribution of CDK1/2 activity to the cytoplasmic localisation of CycA2-eYFP specifically in the G2 phase, we followed individual cells that had a low but clear presence of CycA2-eYFP in the cytoplasm at the time of addition of inhibitors. To improve comparison, we synchronised these cells in silico at the time point when each cell reaches a certain level of cytoplasmic CycA2-eYFP (Fig S3A). We did not observe significant differences in the dynamics of cytoplasmic accumulation of CycA2-eYFP in either of the treatments, showing that a reduction of CDK1/2 activity does not compromise cytoplasmic accumulation of CycA2-eYFP once initiated (Fig S3A).

To test if CDK1/2 activity could promote the onset of cytoplasmic accumulation of CycA2-eYFP, we increased CDK activity using a Wee1 inhibitor. Wee1 inhibition increased the amount of mitotic cells and decreased the duration between cytoplasmic appearance of CycA2-eYFP and mitotic entry, suggesting that CDK activity is increased and that G2 phase is therefore shortened. However, we did not detect an increased rate of cytoplasmic appearance of CycA2-eYFP after Wee1 inhibition (Fig S3B). Thus, we find no evidence that CDK1/2 activity regulates CycA2 appearance in the cytoplasm at the S/G2 transition.

### Lack of CDK1 can lead to loss of cytoplasmic CycA2

Given the lack of evidence for key G2 kinase activities to modulate CycA2 localisation, we reasoned that perhaps a change in binding partner could explain CycA2 cytoplasmic localisation. CycA2 is described to complex predominately with CDK2 in the S phase and increasingly with CDK1 as cells approach mitosis (Merrick et al, 2008). Furthermore, whereas CDK2 is mostly nuclear, CDK1 is present both in the nucleus and in the cytoplasm (Pines & Hunter, 1991, 1994; Moore et al, 1999), therefore potentially providing a mechanism to regulate CycA2 localisation.

To investigate the involvement of CDK–Cyclin complex formation in the localisation of CycA2 we used siRNAs to target either CDK1 or CDK2 (Fig S3C). Live-cell imaging of RPE CycA2-eYFP cells after CDK1 or CDK2 knockdown revealed reduced numbers of cells going through mitosis. Furthermore, CDK1 knockdown increased mitotic duration, showing that knockdown of either CDK affected cell cycle progression (Fig S3D). Analysis of quantitative immunofluorescence in single cells revealed that knockdown of CDK1 led to an increase in the number of cells in G2, presumably because of the lengthening of G2 phase, and subsequently, to the amount of cells with

cytoplasmic CycA2 (Fig 4A and B). Interestingly, a subset of G2 cells contained high nuclear CycA2 and low cytoplasmic CycA2 levels after CDK1 siRNA, indicating that CDK1 may facilitate the localisation of CycA2 to the cytoplasm (Fig 4A, arrows; Fig 4B, grey triangle). On the other hand, CDK2 knockdown led to a marked decrease in number of cells in G2 phase, explaining the reduced level of CycA2 and CDK1 at a population level (Figs 4A and B and S3C). Contrary to the observation after CDK1 depletion, the relation between nuclear and cytoplasmic CycA2 was similar after CDK2 and control knockdown (Fig 4B). Given that knockdown of CDK1 decreased cytoplasmic Cyclin A2 levels, we sought to test if Cdk1 binds to cytoplasmic CycA2 and CDK2 binds to nuclear CycA2. However, we find that both CDK1 and CDK2 are present in both nuclear and cytoplasmic CycA2 and CycA2-eYFP immunoprecipitates (Fig 4C). Thus, our data show that depletion of CDK1 can impair the cytoplasmic localisation of CycA2 in G2 phase, but we cannot exclude the possibility that CycA2 localisation is affected by a stress response associated with depletion of a key cell cycle regulator (see below). As cytoplasmic CycA2 exists in complex with both CDK1 and CDK2, we conclude that the cytoplasmic appearance of CycA2 cannot be explained solely by association with CDK1.

### CycA2 associates with chromatin during the S phase

We next sought to test if CycA2 is restricted to the nucleus during the S phase. The defining process of the S phase is replication of DNA, and Cyclin A2 has been reported to associate directly with the replication protein PCNA (Cardoso et al, 1993; Koundrioukoff et al, 2000; Frouin et al, 2005). We therefore synchronised RPE-CycA2-eYFP cells by addition of hydroxyurea (HU) and assessed the chromatin association of CycA2-eYFP using Chromatin immuno-precipitation (ChIP) followed by qPCR at five chromosomal locations: two expressed promoter regions (A, MYC and B, IRF1), one origin of replication (C, LAMB2 [Abdurashidova et al, 2000]), the α-satellite region (D) and an intragenic region on Chromosome 1 (E). We compared the chromatin binding pattern of CycA2-eYFP at early replicated regions (i.e., expressed genes) and regions undergoing late replication (gene desert areas). When cells approached the mid–S phase 4 h after HU washout we detected an increase in CycA2-eYFP chromatin binding at the expressed genomic locations (A, B and C in Fig 5A), whereas we did not find a significant increase at the α-satellite and intragenic regions. Interestingly, the chromatin association of CycA2-eYFP decreased 8 h after release from HU, at a time when many of the cells had completed DNA replication (Fig 5A). This suggests that CycA2-eYFP association with chromatin is increased during ongoing DNA replication. We therefore sought to test whether CycA2-eYFP chromatin association restricted CycA2-eYFP mobility in the nucleus. To this end, we used FRAP, and compared the nuclear recovery of RPE-CycA2-eYFP cells expressing CycA2-eYFP only in the nucleus (S phase) or in both nucleus and cytoplasm (G2 phase). To avoid overstating the immobile fraction

---

anti-CycA2 (E) or anti-PLK1 (F) antibodies and bound proteins were probed with indicated antibodies. For (E), data using additional antibodies can be found in Fig 4C.
**(G)** Expression of PLK1 T210D can partially rescue CycA2 siRNA-mediated cell cycle arrest. U2TR cells with inducible expression of PLK1 T210D were transfected with CycA2 siRNA. 20 h later, Doxycycline was added and cells were followed using phase contrast time-lapse microscopy. 300 cells per condition were followed manually over time. All experiments were repeated at least three times.

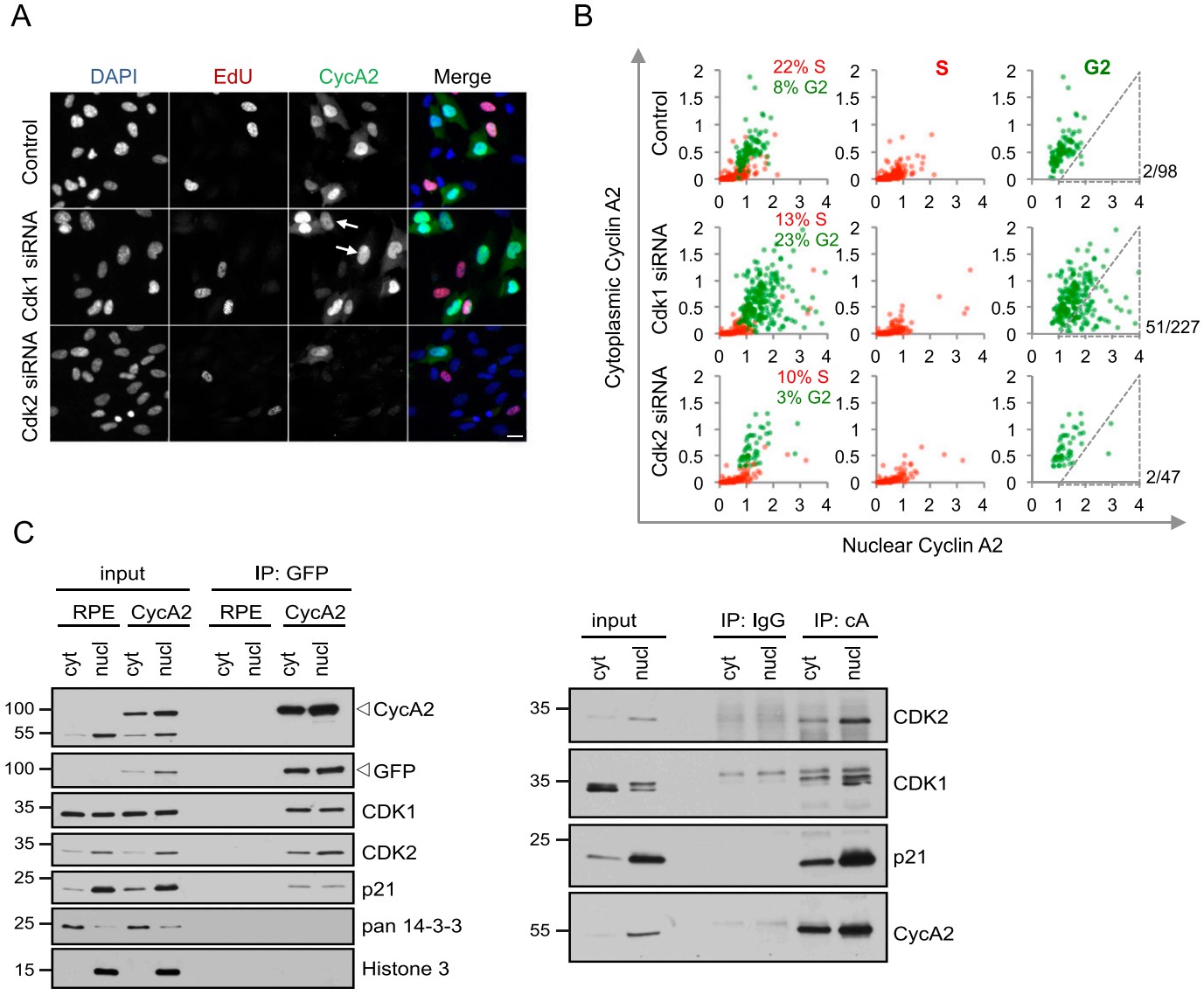

**Figure 4. CDK1 can contribute to cytoplasmic accumulation of Cyclin A2.**
**(A)** RPE cells were transfected with siRNAs for either CDK1 or CDK2 for 48 h, incubated with EdU for 20 min and fixed. Arrows indicate G2 cells with low cytoplasmic CycA2. Scale bar 20 $\mu$m. **(B)** Quantification of cytoplasmic and nuclear integrated intensities of CycA2 in at least 500 RPE cells imaged as in (A). Cells were gated for DAPI and EdU levels and assigned to the S phase (red dots) or G2 phase (green dots). Each dot represents one cell; the percentages indicate the proportion of the S- and G2 phase cells in each condition. Number of cells within indicated gate and total amount of G2 cells are shown to the right. **(C)** RPE (RPE) or RPE CycA2-eYFP (CycA) cells were synchronised in G2, separated to cytosolic and nuclear fractions and immunoprecipitated with GFP Trap (left) or with control IgG and CycA2 antibody (right). Proteins bound to the carrier were probed with indicated antibodies. For (C), data using additional antibodies can be found in Fig 3E. All experiments were repeated at least three times.

due to bleaching and focus changes, we normalised the bleached region to an unbleached area in the same nucleus. Although the recovery of fluorescence was somewhat slower in the S phase versus in the G2 phase, we found no major difference between the recovery curves, suggesting that the mobility of CycA2-eYFP is not dramatically altered between S and G2 phases. Moreover, in contrast to PCNA that shows a large immobile fraction in replication foci (Trembecka-Lucas et al, 2013), the immobile component of CycA2-eYFP is limited during both the S- and G2 phases (Fig 5B). This suggests that a majority of CycA2 is not stably bound to chromatin. Indeed, using

protein fractionation instead of ChIP, we do not detect a marked decrease of CycA2 in the insoluble—chromatin bound—fraction at late times after release from HU. Rather, whereas both CycA2 and CycA2-eYFP increase in the cytoplasmic fraction, their component in the insoluble fractions remains unaltered (Fig 5C). Taken together, this indicates that CycA2-eYFP associates more with replicating chromatin, but also that CycA2 cytoplasmic appearance at the S/G2 transition cannot be explained by a decrease in chromatin association. Rather, CycA2 chromatin association remains similar as total and cytoplasmic CycA2 levels increase through the G2 phase.

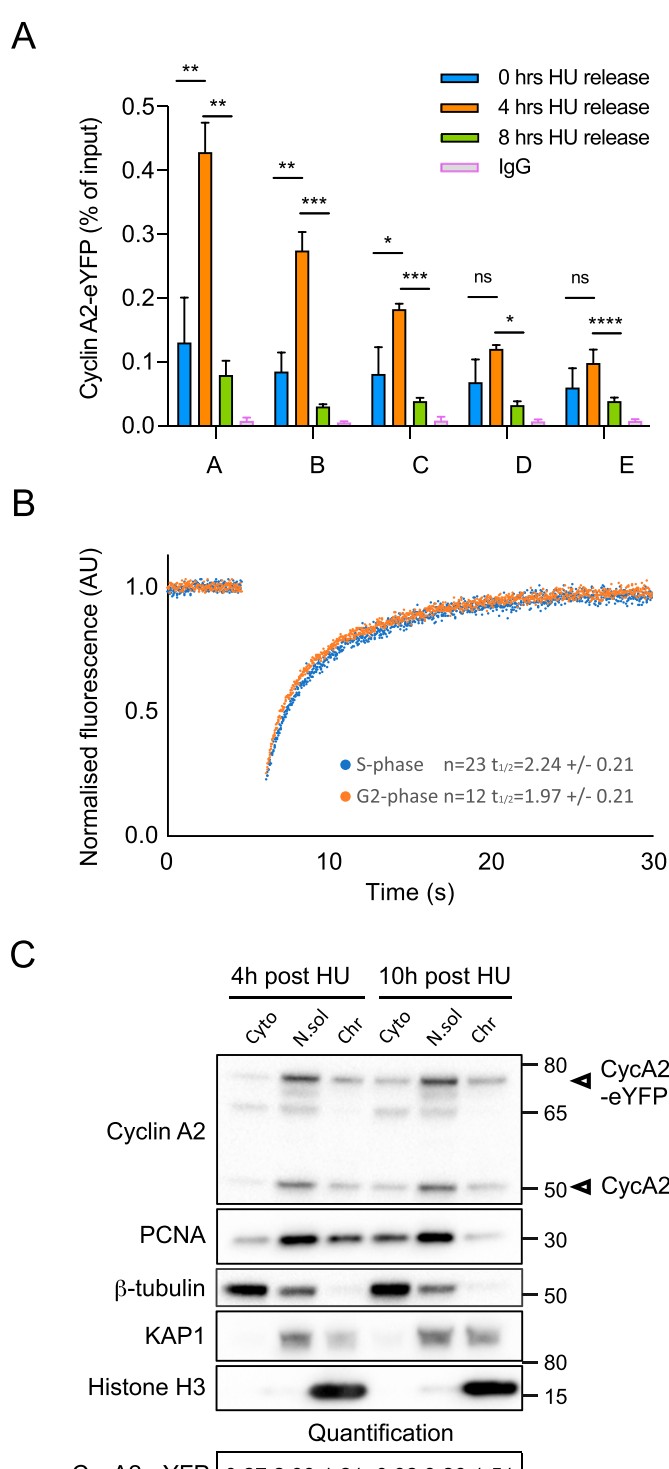

**Figure 5. Cyclin A2 associates with chromatin during the S phase.**
**(A)** Chromatin immunoprecipitation (ChIP) in RPE-CycA2-eYFP cells, synchronised with hydroxyurea (HU, 2 mM in PBS) for 18 h and harvested after HU washout at different time points (0, 4 and 8 h). The enrichment of CycA2-eYFP and IgG, determined by ChIP-qPCR is plotted as percentage of input (ratio of immunoprecipitated DNA to the total amount of DNA). Genomic locations are indicated by alphabetical letters A–E (details in Table S1). Graph shows one representative experiment. Mean value and SD are based on duplicate

### The DNA damage response modulates cytoplasmic accumulation of CycA2 through p21

DNA replication functions as a brake on cell cycle progression by limiting activities that promote mitosis (Lemmens et al, 2018; Saldivar et al, 2018). A similar limitation of activities that promote mitosis is achieved by DNA damage. At least for certain signalling components, the S phase resembles a low-level DNA damage response (Sørensen & Syljuåsen, 2012; Lemmens & Lindqvist, 2019). We therefore sought to test whether the occurrence of cytoplasmic CycA2 is affected by DNA damage. After addition of the radiomimetic drug Neocarzinostatin (NCS) or of the Topoisomerase II inhibitor etoposide, cytoplasmic CycA2-eYFP signal disappeared (Figs 6A–C and S4A). The reduction in cytoplasmic CycA2-eYFP occurred between 2 and 5 h after NCS or etoposide addition and was mirrored by a rapid increase in nuclear fluorescence. Such an increase in nuclear fluorescence was only detected in cells containing cytoplasmic CycA2-eYFP (Fig 6B). Furthermore, the loss of cytoplasmic fluorescence and the increase in nuclear fluorescence was not affected by addition of a proteasome inhibitor (Fig 6C), indicating that upon DNA damage, cytoplasmic Cyclin A2 translocates to the nucleus. As a consequence, the nuclear fluorescence in G2 cells exposed to NCS or etoposide exceeded the nuclear fluorescence observed during an unperturbed G2 phase (Fig 6C).

A few hours after loss of CycA2-eYFP in the cytoplasm, CycA2-eYFP signal disappeared also from the nucleus (Fig 6A–C). In contrast to cytoplasmic CycA2-eYFP signal, nuclear CycA2-eYFP signal remained after addition of a proteasome inhibitor, suggesting that nuclear CycA2-eYFP loss involves protein degradation (Fig 6C). The loss of cytoplasmic CycA2-eYFP occurred at similar time-scales after DNA damage as what we and others previously described for p53- and p21-dependent nuclear translocation of CycB1-eYFP (Mullers et al, 2014; Krenning et al, 2014; Johmura et al, 2014). Moreover, loss of nuclear CycA2-eYFP was apparent in cells receiving DNA damage in G2 phase, but was rarely visible in cells targeted in the S phase, which would be consistent with a role for p21, whose expression is suppressed during the S phase despite the presence of damaged DNA (Fig 6A and B) (Müllers et al, 2017; Sheng et al, 2019). To test whether p53-p21 affects CycA2 localisation, we transfected cells with either p21 or p53 siRNAs for 48 h and assessed the dynamics of CycA2-eYFP upon DNA damage. Interestingly, p21 and p53 knockdown impaired both the cytoplasmic and subsequent nuclear loss of CycA2-eYFP after addition of etoposide (Fig 6D). This suggests that p21 and p53 can modulate both localisation and levels of CycA2 after DNA damage.

samples, each with duplicate qPCR reactions. Mean relative recovery for non-immune IgG are 0.0072. Asterisks indicate two-tailed paired *t* test; *$P < 0.05$, **$P < 0.01$, ***$P < 0.001$, ****$P < 0.0001$. **(B)** FRAP in the nucleus of RPE-CycA2-eYFP cells. Graph shows normalised average of cells with (G2 phase) and without (S phase) detectable cytoplasmic CycA2-eYFP. N = 2. **(C)** RPE-CycA2-eYFP cells were synchronised with hydroxyurea for 18 h and harvested after 4 and 10 h in HU-free media. Cytoplasmic proteins (Cyto), soluble nuclear proteins (N.sol), and chromatin bound proteins (Chr) were isolated and detection of Cyclin A2 and PCNA was performed by Western blot. 5 μg of protein was loaded for each fraction. β-Tubulin, KAP1, and Histone H3 were used as controls of cytosolic, nuclear soluble and chromatin fraction, respectively. N = 2.

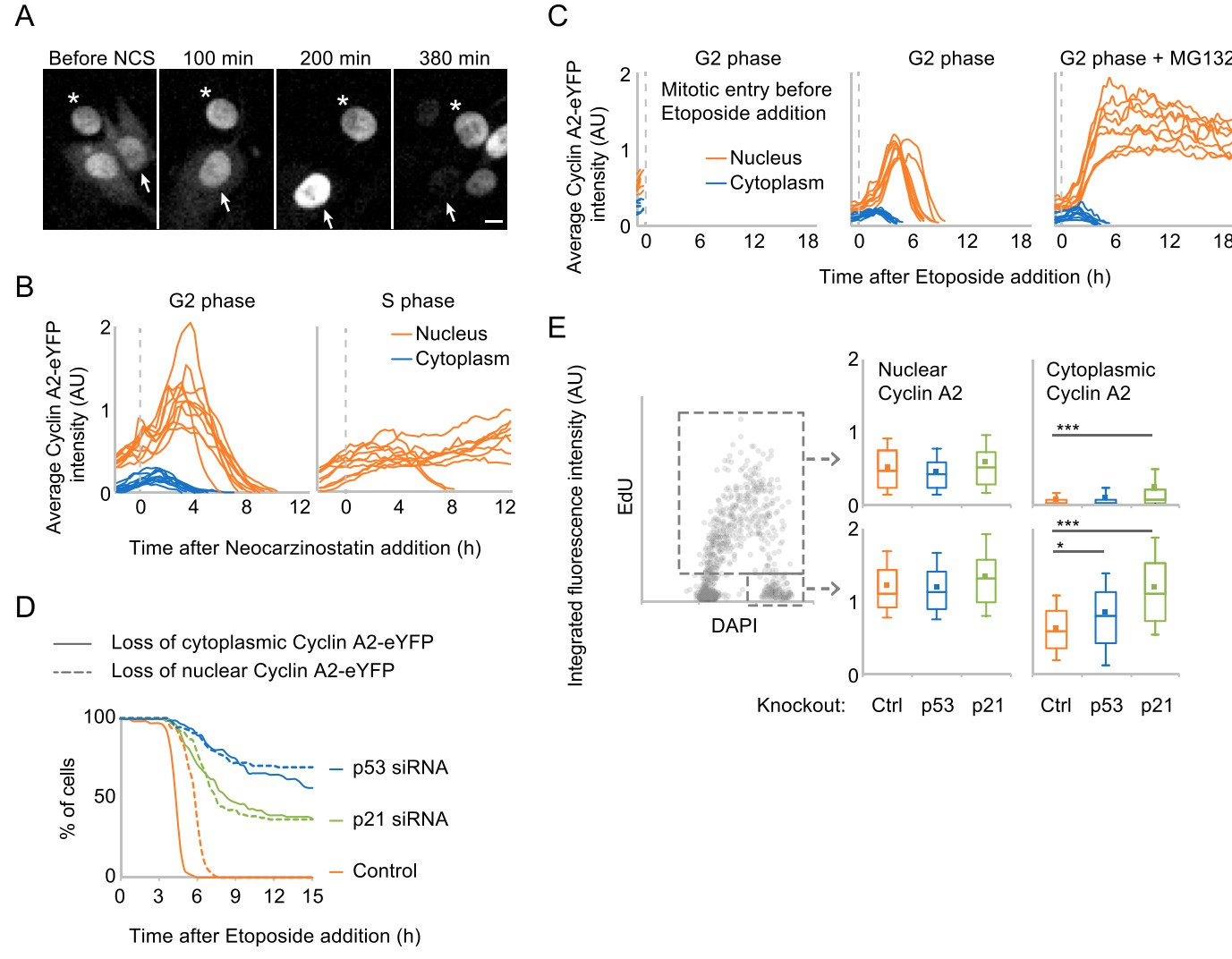

**Figure 6. p21 can modulate cytoplasmic accumulation of Cyclin A2-eYFP.**
**(A)** Time-lapse images of RPE CycA2-eYFP cells treated with 2 nM Neocarzinostatin (NCS). Asterisk indicates a cell exposed to NCS in the S phase, and arrow indicates a cell exposed to NCS in the G2 phase. Scale bar 10 $\mu m$. **(B)** Quantification of average nuclear and cytoplasmic CycA2-eYFP intensity after NCS addition. RPE CycA2-eYFP cells were imaged as in (A). Graphs show quantification over time in single cells, separated in cells showing cytoplasmic CycA2-eYFP at time of NCS addition (left, G2 phase, n = 10) or not (right, S phase, n = 10). No cell entered mitosis. **(C)** Quantification of average nuclear and cytoplasmic CycA2-eYFP intensity after etoposide and MG132 addition. RPE CycA2-eYFP cells that showed both nuclear and cytoplasmic fluorescence (G2 cells) at start of experiment were followed over time. Graphs show quantification over time in single cells, separated in cells entering mitosis before etoposide addition (left, n = 9, plotted until time point before mitosis), etoposide addition alone (middle, n = 8), etoposide, and MG132 addition (n = 9). **(D)** RPE CycA2-eYFP cells were transfected with p21, p53 or control siRNAs for 48 h and treated with etoposide at t = 0. Single cells were tracked over time and the time point of loss of CycA2-eYFP was determined visually. Graph shows cytoplasmic and nuclear loss of CycA2-eYFP of at least 100 cells that contained cytoplasmic CycA2-eYFP at time point of etoposide addition. **(E)** WT (ctrl), p21$^{-/-}$ or p53$^{-/-}$ RPE cells were incubated for 20 min with EdU and fixed. Graph to left shows quantification of integrated intensity of EdU staining versus nuclear DAPI intensity in at least 1,500 wt cells; each circle represents one cell. The large grey rectangle indicates gating for EdU positive cells (S phase) and the small grey rectangle indicates gating for EdU-negative 4N cells (G2). Box plots to right show 90, 75, 50, 25, and 10 percentiles of cells in S phase (top) and in G2 phase (bottom). Squares indicate average value. * indicates $P < 0.05$, *** indicates $P < 0.0001$, $t$ test.

We detected an interaction between p21 and CycA2 in the absence of externally induced DNA damage. The interaction was detected in both cytoplasmic and nuclear fractions, raising the possibility that association with p21, which contains an NLS (Rodríguez-Vilarrupla et al, 2002), can help translocate CycA2 to the nucleus (Fig 4C). To test whether p53 and p21 contribute to the regulation of CycA2 localisation in the absence of DNA damage, we next used CRISPR/Cas9 to establish p21 or p53 deficient RPE cell lines (Fig S4B). We fixed p21$^{-/-}$, p53$^{-/-}$, and WT RPE cells after a short pulse with EdU, stained using DAPI and antibodies against CycA2, and quantified the levels of nuclear and cytoplasmic CycA2 in S or G2 phase (Fig 6E). We observed that both p21$^{-/-}$ and p53$^{-/-}$ cell lines accumulated nuclear CycA2 similar to the parental cell line. However, quantification of cytoplasmic CycA2 revealed that p21 deficient cells showed an increase in cytoplasmic CycA2, most apparent in the G2 phase. To a lower extent, we also detect increased cytoplasmic CycA2 staining in p53$^{-/-}$ G2 cells, but not in S phase cells (Fig 6E). Taken together, our data indicate that p21 contributes to keep CycA2 nuclear, but do not support that p21 is a determining factor behind the appearance of cytoplasmic CycA2 at the S/G2 transition. However, as

p21 levels increase after activation of a DNA damage response, p21 efficiently inhibits cytoplasmic localisation of CycA2.

## Discussion

The S/G2 transition is marked by an increase in CDK1 and PLK1 activities, which through multiple feedback loops slowly build up until enforcing mitotic entry (Akopyan et al, 2014; Crncec & Hochegger, 2019). How to separate the hen from the egg and identify a starting point in these feedback loops has remained an unsolved question, but several studies have suggested a role for CycA2 as an initiating activity (Mitra & Enders, 2004; Fung et al, 2007; Gong et al, 2007; De Boer et al, 2008; Gheghiani et al, 2017; Vigneron et al, 2018). However, CycA2-CDK2 activity is already present during the S phase, raising the question why PLK1 activation is detected only after completion of DNA replication. One part of the answer is that DNA replication activates ATR-CHK1, which restricts mitotic kinases. However, inhibition of ATR-CHK1 does not cause immediate activation of PLK1 in all S-phase cells, suggesting that other mechanisms contribute to restrict PLK1 activation until completion of the S phase (Lemmens et al, 2018). Interestingly, whereas proteins as PLK1 and CycB1-CDK1 are both nuclear and cytoplasmic, Bora—a node in the feedback loops coupling Aurora A, CDK, and PLK1—appears exclusively cytoplasmic (Feine et al, 2014; Bruinsma et al, 2015).

Here we show that CycA2 appears in the cytoplasm at the S/G2 transition. We propose a model in which the rising cytoplasmic activity of CycA2-CDK initiates activation of PLK1 through phosphorylation of the cytoplasmic cofactor Bora (Fig 7). Later in G2, combined activities of CycA2-CDK and CycB-CDK1 can further increase activation of PLK1 through massive modification of Bora, eventually resulting in commitment to mitosis and protection of Bora from SCF-dependent degradation (Feine et al, 2014; Tavernier et al, 2015; Thomas et al, 2016; Gheghiani et al, 2017; Vigneron et al, 2018). We find no evidence that CDK1 or PLK1 activities influence CycA2 localisation, supporting the idea that rather than a component of feedback loops, CycA2 appearance in the cytoplasm functions as a trigger for mitotic kinase activation. A parallel nuclear pathway for CDK-mediated activation of PLK1 by WAC has been described (Qi et al, 2018). It remains unclear how nuclear activation of PLK1 fits with our observation that cytoplasmic but not nuclear CycA2 associates with PLK1, and the observation that PLK1 restricted to the nucleus is not activated (Bruinsma et al, 2015). One

alternative can be that after initial activation mediated through Bora in the cytoplasm, additional mechanisms promote PLK1 activation at later stages in the G2 phase. Nonetheless, PLK1 contains an NLS, which at least in *Drosophila* is exposed upon activation, allowing re-distribution of active PLK1 in the cell (Taniguchi et al, 2002; Kachaner et al, 2017). It would be interesting to follow if the spatial and temporal pattern of PLK1 activation differs between expression of wild-type and solely cytoplasmic CycA2.

The main mechanism that regulates CycA2 cytoplasmic appearance at the S/G2 transition remains unclear, although we have identified contributing factors. First, CycA2 can associate with replicating chromatin, which can assist in keeping CycA2 nuclear during the S phase. The restriction to the nucleus continues at least to some level during the G2 phase, and we find that a main player in this process is the stochiometric Cyclin/CDK interactor p21. A basal level of p21 expression occurs independently of p53 in the absence of induced DNA damage, although p21 levels in the S phase are generally low due to DNA replication–dependent degradation (Macleod et al, 1995; Kim et al, 2008; Nishitani et al, 2008). The basal level of p21 can be enhanced in a p53-dependent manner, which can restrict cytoplasmic appearance of CycA2 in case of replication errors or other cellular stresses. Experimental interference with the cell cycle can cause stress and p53 induction, for example, as a consequence of mitotic delay (Uetake & Sluder, 2010). Caution is therefore required when interpreting results after cell cycle perturbation, such as depletion of CDK1. In addition, absence of negative regulators of CDK activity as p53 and p21 could have secondary consequences due to enhanced CDK activity.

The levels of CycA2 increase dramatically from the early S phase to late G2 phase (Fig 1C). An attractive but speculative model would be that DNA replication and non-DNA replication-dependent interactions keep an amount of Cyclin A2 sequestered in the nucleus. This raises the possibility that increasing CycA2 levels through G2 phase could eventually surpass the amount of CycA2 sequestered in the nucleus and appear in the cytoplasm. How additional mechanisms behind CycA2 cytoplasmic appearance that are not identified in this study fit into the model remains unclear (Fig 7).

A long standing idea in the cell cycle field has been that upon stress, cell cycle progression can be delayed by altering the localisation of key proteins as Cdc25B, Cdc25C, and CycB1 (Takizawa & Morgan, 2000). We and others previously showed that upon DNA damage, terminal cell cycle exit from the G2 phase is marked by a p21-dependent abrupt translocation of CycB1 to the nucleus, followed by APC/C-dependent degradation (Johmura et al, 2014;

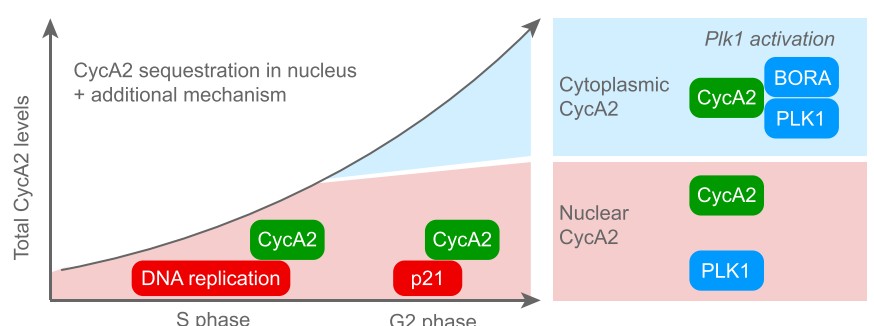

**Figure 7. A model for polo-like kinase 1 activation by cytoplasmic CycA2.**
CycA2 is restricted to the nucleus during the S phase. At the S/G2 transition, part of CycA2 localises to the cytoplasm, where it can phosphorylate Bora, resulting in activation of polo-like kinase 1.

Krenning et al, 2014; Mullers et al, 2014). Here, we find that with respect to pattern of expression, localisation changes, and timings, CycA2 behaves similar to CycB1 after DNA damage in the G2 phase. It is tempting to speculate that CycA2 nuclear restriction after DNA damage prevents PLK1 activation in the cytoplasm, while allowing nuclear functions such as stimulating transcription of p21 (Müllers et al, 2017). This opens up the possibility that CycA2 not only is upstream in activating mitotic kinases, but also for triggering cell cycle exit from the G2 phase in case of excessive DNA damage.

The regulated appearance of CycA2 in the cytoplasm at the S/G2 transition suggests that CycA2 can direct CDK activity both in a temporal and spatial manner. This is similar to CycB1, whose change in localisation before mitosis provides access to nuclear substrates (Pines & Hunter, 1991). Given that multiple cytoplasmic targets of CycA2 have been described, and that CDK2 is identified as a major interaction hub in the cytoskeleton, we note the possibility that apart from PLK1 activation, various processes are differentially regulated before and after completion of the S phase (Woodman et al, 1993; Tsang et al, 2007; Arsic et al, 2012; Kanakkanthara et al, 2016; Thul et al, 2017).

## Materials and Methods

### Cell culture

Human hTERT-RPE1 (hereafter referred to as RPE), U2OS and HeLa cells were cultured in an ambient-controlled incubator at 37°C and 5% $CO_2$. All cells were a kind gift from René Medema and were regularly controlled for mycoplasma infection. RPE cells were cultured using DMEM-F12 + GlutaMAX (Invitrogen) supplemented with 10% heat-inactivated FBS (HyClone) and 1% P/S (HyClone). U2OS and HeLa cells were cultured using DMEM + GlutaMAX (Invitrogen) supplemented with 6% heat-inactivated FBS (HyClone) and 1% Penicillin/Streptomycin (P/S; HyClone). For adeno-associated virus production, HEK293 cells were cultured using DMEM + GlutaMAX (Invitrogen) supplemented with 10% heat-inactivated FBS (HyClone) and 1% Penicillin/Streptomycin (P/S; HyClone). For live-cell imaging experiments the medium of the cells was changed to Leibowitz-15 (Invitrogen) supplemented with 10% FBS (HyClone) and 1% P/S (HyClone).

### Establishment of cell lines

RPE CycA2-eYFP cells were obtained by adeno-associated (AAV)–mediated homologous recombination as previously described (Akopyan et al, 2014). Briefly, the targeting cassette was designed to contain an arm of 1.1 kb of homology with the sequence directly 5′ of the CCNA2 Stop codon followed by the ORF of EYFP and another arm of homology 1.016 kb with the 3′ UTR of the CCNA2 gene. Adeno-associated viruses containing the homology cassette were produced and used to transduce RPE cells. 4 d after transduction cells were sorted by FACS to enrich the YFP-positive population. After two rounds of sorting, single YFP-positive cells were seeded in 96-well plates and clones which were validated by Western blot and live-cell imaging. Knockout of TP53 and p21 in RPE cells using CRISPR/Cas9 was generated as described previously (Pechackova et al, 2016) and independent clones were validated by Western blotting

and sequencing of the genetic loci. For expression of CycA2 localisation variants, CycA2 constructs containing silent mutations to the four SMARTpool ON-TARGET plus siRNAs targeting CycA2, linked to mCherry and coupled to NLS or NES sequences as indicated in Fig 3C, and flanked by Piggy-Bac integration sites, were synthesised by Vector Builder. Sequence information is available at https://en.vectorbuilder.com/design/retrieve.html for vectors VB190920-1050khh and VB190920-1053jda. The vectors were co-transfected into RPE p53$^{-/-}$ cells expressing Dox-inducible OsTIR from the Rosa locus (Lemmens et al, 2018), with vectors for GFP flanked by Piggy-Bac integration sites and Piggy-Bac transposase. After selection with blasticidin, polyclonal populations were used and only GFP-positive cells were analysed.

### Cell synchronisation

RPE or RPE CycA2-eYFP cells were synchronised in G0 by growing to confluency, split to fresh medium supplemented with thymidine (2 mM) and grown for 40 h. Cells were released to fresh medium and collected after 5 h. Synchronisation efficiency was validated by flow cytometry using 4n DNA content and absence of pS10-H3 staining as G2 markers. Typically, this protocol yielded >95% G2 population and less than 0.5% mitotic cells. Alternatively, where indicated, RPE or RPE CycA2-eYFP cells were synchronised in the S phase by addition of HU.

### Cell fractionation and immunoprecipitation

Cells were fractionated using hypotonic lysis as previously described (Andersen et al, 2002). Briefly, cells were collected by trypsinization and centrifugation (300g for 5 min at 4°C), and washed with PBS. The cell pellet was resuspended in 5× packed cell volume of hypotonic buffer A (10 mM HEPES-KOH, pH 7.9, 10 mM KCl, 1.5 mM $MgCl_2$, 0.5 mM DTT, and 0.5 mM PMSF) supplemented with a cocktail of protease inhibitors (cOmplete, EDTA free; Roche) and phosphatase inhibitors (PhosSTOP; Roche) and incubated on ice for 5 min. Next, the cells were spun down at 500g for 5 min, suspended in 2× packed cell volume of supplemented buffer A and dounced using a tight-fitting pestle. Nuclei were collected by centrifugation at 500g for 5 min at 4°C. Supernatant was centrifuged 20,000g 10 min 4°C, supplemented with NaCl and Triton X-100 to 150 mM and 0.1% final concentration, respectively, and used as cytoplasmic fraction. Nuclei were cleaned by centrifugation over sucrose gradient, lysed in lysis buffer (10 mM Hepes, pH 7.9, 10 mM KCl, 150 mM NaCl, 1.5 mM $MgCl_2$, 0.1% NP-40, 0.5 mM DTT, 0.5 mM PMSF supplemented with protease inhibitor cocktail, and PhosSTOP), sonicated and cleared by centrifugation at 20,000g 10 min 4°C. For IP 2 mg of cytoplasmic and nuclear extracts were incubated with either 15 µl GFP Trap beads (ChromoTek) for 1 h at 4°C or 1–1.5 µg IgG overnight at 4°C, Protein A/G Ultralink beads were added for last 2 h. Beads were washed four times with lysis buffer and precipitates were eluted to SDS–PAGE sample buffer.

For Fig 5C, protein fractionation was performed according to Baranello et al (2016) with minor changes. To prepare the extracts, cells were washed once with PBS and twice with Hepes wash buffer (20 mM Hepes, pH 7.5, 137 mM NaCl). After incubation with Hepes lysis buffer (10 mM Hepes, pH 7.5) cell were scraped and collected, smashed through a 20 gauge needle and then centrifuged for 10 min at 800g at 4°C. Supernatant containing the cytoplasmic

proteins was collected and the pellet was washed with Hepes lysis buffer, centrifuged and finally resuspended in RIPA 137 buffer (50 mM Tris–HCl, pH 7.5, 137 mM NaCl, 1% NP-40). A 25-gauge needle was used to destroy the nuclear membranes, then the sample was centrifuged at 1,500$g$ for 10 min 4°C and the supernatant containing nuclear soluble proteins was collected. Pellet was resuspended in RIPA 400 buffer (50 mM Tris–HCl, pH 7.5, 400 mM NaCl, 1% NP-40) and sonicated with Diagenode Bioruptor Sonicator for 5 min (low sonication setting, 30 s on–30 s off). Samples were centrifuged at 13,500$g$ for 10 min at 4°C and supernatant containing chromatin bound nuclear proteins collected and equilibrated to RIPA 137 buffer.

### Chromatin immunoprecipitation (ChIP) and qPCR

ChIP samples were prepared from RPE-CycA2-eYFP cells as described in Baranello et al (2016) with minor changes. Briefly, 8 × 10⁶ cells were cross-linked with 1% formaldehyde for 5 min. Cross-linking was stopped by the addition of glycine to 125 mM final concentration and cells were washed twice with cold PBS. After harvesting cells by scraping, the pellet was washed once with PBS plus 0.5% BSA and resuspended in TE-SDS 0.1% (10 mM Tris–HCl, pH 8.0; 1 mM EDTA, pH 8.0, SDS 0.1%) supplemented with complete protease inhibitor tablet (Roche) to a final concentration of 4.5 × 10⁶ cells/ml. Samples were sonicated for 20 min with Covaris ME220 sonicator using the 1-ml High Cell protocol (Peak power: 75, Duty % factor: 15, Cycles/Burst: 1,000, Average power: 11.25) to produce chromatin fragments of 400 bp on average. After clarification by centrifugation, sonicated extracts were adjusted to the conditions of RIPA buffer (10 mM Tris, pH 8.0, 1 mM EDTA, pH 8.0, 1% Triton X-100, 0.1% SDS, 200 mM NaCl, Na-deoxycholate 0.1%).

3 $\mu$g of anti-GFP (ab290) or IgG (sc2025) were mixed with 40 $\mu$l of Dynabeads Protein A (Invitrogen) and incubated at 4°C for 6 h with rotation. Chromatin from 8 × 10⁶ cells was added to the protein A–antibody complexes and incubated overnight at 4°C with rotation. Immunoprecipitates were washed twice with RIPA buffer (10 mM Tris–HCl, pH 8.0, 1 mM EDTA, pH 8.0, 1% Triton X-100, 0.1% Na-deoxycholate, 0.1% SDS, and 200 mM NaCl); twice with RIPA buffer plus 300 mM NaCl; once with LiCl buffer (10 mM Tris–HCl, pH 8.0, 1 mM EDTA, pH 8.0, 250 mM LiCl, 0.5% NP-40, 0.5% Na-deoxycholate); and twice with TE. The beads were then resuspended in 100 $\mu$l TE plus 0.25% SDS supplemented with proteinase K (500 $\mu$g/ml; Roche) and incubated overnight at 60°C. The DNA was recovered from the eluate by phenol chloroform extraction followed by ethanol precipitation in the presence of 20 $\mu$g of GlycoBlue (Invitrogen) and dissolved in TE. qPCR was performed using the 7500 Fast Real-Time PCR system (Applied Biosystem) and Fast SYBR Green Master Mix (Thermo Fisher Scientific). 7500 Software (Applied Biosystem) was used for quantification by the standard curve method and data was plotted as percentage of input, ratio of the signal from immunoprecipitated DNA to the signal of input DNA. Five genomic locations were analysed. The detection primers are listed in Table S1.

### Inhibitors

For live-cell imaging and quantitative immunofluorescence experiments, the following inhibitors were used at the indicated concentrations for 4 h unless indicated differently in the experiments: RO-3306 at 10 $\mu$M (CDK1 inhibitor; Calbiochem), NU6140 at 10 $\mu$M (CDK2 inhibitor; Calbiochem), MK-1775 at 5 $\mu$M (Wee1 inhibitor; Selleck Chemicals), etoposide 2 $\mu$M (topoisomerase II inhibitor; Sigma-Aldrich), neocarzinostatin at 2 nM (radiomimetic drug; Sigma-Aldrich), thymidine at 2.5 mM (Sigma-Aldrich), and hydroxyurea at 2 mM (ribonucleotide reductase inhibitor; Sigma-Aldrich).

### siRNA transfection

SMARTpool ON-TARGET plus siRNAs targeting CycA2, CDK1, CDK2, p21 or p53 as well as a scrambled control siRNA were purchased from Dharmacon and used at a concentration of 20 nM using HiPerFect transfection reagent (QIAGEN) and OptiMEM (Invitrogen) at 48 and 24 h before live-cell imaging or fixation.

### Live-cell microscopy

For live-cell imaging, cells were seeded in 96-well imaging plates (BD Falcon) using Leiboiwitz-15 medium (Invitrogen) and followed on an ImageXpress system (Molecular Devices) using a 20× NA 0.45 objective or on a Leica DMI6000 Imaging System using a 20× NA 0.4 objective. Images were processed and analysed using ImageJ. Nuclei and cytoplasm were selected by manual drawing. FRET microscopy was performed as in Hukasova et al (2012) and simultaneous monitoring of FRET and a PCNA chromobody was performed as in Akopyan et al (2014).

### FRAP

FRAP microscopy on RPE CycA2-eYFP was performed on a Zeiss LSM710 microscope, using a 40×/NA1,2 objective and a 514 nm laser. Fluorescence intensities of two 10 × 50 pixel regions in the same nucleus were recorded at 44 images/second. Bleaching was performed in one region with 100% laser power for 1.5 s after acquiring 200 images. After background subtraction, estimated from an area next to the cell before scanning, the ratio between the bleached and unbleached region in the same nucleus was calculated over time.

### Antibodies

The following antibodies were used: GFP (ab13970; Abcam), GFP (ChIP; ab290) CycA2 (#sc-751; Santa Cruz) (used for immunofluorescence), CycA2 (#4656; Cell Signalling), CycA2 (HPA020626), PLK1 (ab14210; Abcam), affinity purified mouse anti pT210-PLK1 (clone K50-483; Becton Dickinson), affinity purified rabbit anti-Bora (Bruinsma et al, 2014), CDK1 (sc-54; Santa Cruz and #9116; Cell Signalling), CDK2 (sc-163; Santa Cruz and #2564; Cell Signalling), GAPDH (G9545; Sigma-Aldrich), H2B (ab1790; Abcam), $\beta$-Tubulin (#2128S; Cell Signalling), PCNA (HPA030522), KAP1 (A300-274A), H3 (ab1791), Alexa Fluor 488–goat anti-chicken (#A11039; Life Technologies), and Alexa Fluor 647–donkey anti-rabbit (#A31537; Life Technologies).

### Quantitative immunofluorescence

For quantitative immunofluorescence experiments 10,000 cells were seeded 16 h before the different treatments with inhibitors. For siRNA transfections, 5,000 cells were seeded instead. 20 min before fixation EdU (5-ethynyl-2'-deoxyuridine; Molecular Probes) was added in all the experiments. Cells were fixed using 3.7% formaldehyde (Sigma-Aldrich) for 5 min and permeabilised using –20°C methanol (Sigma-Aldrich) for 2 min, blocking was performed using 2% BSA (Sigma-Aldrich) in TBS supplemented with 0.1% Tween 20 (TBS-T). After blocking, the cells were incubated with primary antibodies at 4°C overnight. After washing, cells were incubated with secondary antibodies and DAPI for 1 h at room temperature. Click chemistry was performed after wash of the secondary antibody using a mixture of 100 mM Tris, 1 mM $CuSO_4$, 100 mM ascorbic acid and fluorescent dye (#A10277 and #A10266; Invitrogen) and incubated for 1 h at room temperature. Images were acquired on an ImageXpress system (Molecular Devices) using either a 20× (NA) objective or a 40× NA 0.6 objective. Images were manually screened and processed and analysed using CellProfiler (Carpenter et al, 2006) to identify and measure nuclear and cytoplasmic fluorescence intensity of single cells. Cell cycle stages were determined setting a threshold both on DAPI and EdU levels. For time-estimates based on fixed cells, CycA2 and DAPI measurements were corrected using a background function and 2-4N DNA content cells were sorted based on the 90th percentile CycA2 and DAPI measurement, as described in Akopyan et al (2016).

### In vitro kinase assay

Kinase dead PLK1-K82R, GST-Bora, and GST-Aurora-A were purified from bacteria as described (Macurek et al, 2008) and incubated with CycA2-CDK2 (100 ng/reaction; Biaffin GmbH) in a kinase buffer (25 mM MOPS pH 7.2, 12.5 mM glycerol 2-phosphate, 25 mM $MgCl_2$, 5 mM EGTA, 2 mM EDTA, and 0.25 mM DTT) supplemented with 100 $\mu$M ATP and 5 $\mu$Ci 32P-$\gamma$-ATP at 30°C for 30 min. After separation of proteins by SDS–PAGE, phosphorylation was detected by autoradiography or by pT210-PLK1 antibody.

## Data Availability

Data generated or analysed during this study are included in this published article or available upon request.

## Supplementary Information

## Acknowledgements

We thank the members of Lindqvist and Macurek labs for comments and suggestions, and Patrick von Morgen for technical assistance. This work was supported by grants from the Swedish Research Council, the Swedish Foundation for Strategic Research, and the Swedish Cancer Society to A Lindqvist, Knut and Alice Wallenberg foundation (KAW 2016.0161) and the Swedish Research Council (VR 2016-02610) to L Baranello, and the Czech Science Foundation (17-04742S) to L Macurek.

## Author Contributions

H Silva Cascales: conceptualization, investigation, and writing—original draft.
K Burdova: investigation and writing—review and editing.
A Middleton: investigation and writing—review and editing.
V Kuzin: investigation and writing—review and editing.
E Müllers: conceptualization, investigation, and writing—review and editing.
H Stoy: investigation and writing—review and editing.
L Baranello: supervision, funding acquisition, investigation, and writing—review and editing.
L Macurek: supervision, funding acquisition, investigation, and writing—review and editing.
A Lindqvist: conceptualization, supervision, funding acquisition, investigation, and writing—original draft, review, and editing.

### Conflict of Interest Statement

E Müllers is an employee of AstraZeneca. Other authors declare no competing interests.

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
