## [Reviewer comments · Life Science Alliance]

Life Science Alliance

Cyclin A2 localises in the cytoplasm at the S/G2 transition to activate PLK1

Helena Silva Cascales, Kamila Burdova, Anna Middleton, Vladislav Kuzin, Erik Müllers, Henriette Stoy, Laura Baranello, Libor Macurek, and Arne Lindqvist

DOI: <https://doi.org/10.26508/lsa.202000980>

Corresponding author(s): Arne Lindqvist, Karolinska Institutet

Review Timeline:	Submission Date:	2020-12-08
	Editorial Decision:	2020-12-15
	Revision Received:	2020-12-22
	Accepted:	2020-12-23

Scientific Editor: Shachi Bhatt

Transaction Report:

Please note that the manuscript was reviewed at Review Commons and these reports were taken into account in the decision-making process at Life Science Alliance.

Review
COMMONS

1st Authors' Response to Reviewers

We thank the reviewers for the constructive review. We have added our response point-by-point below. All changes in the manuscript are marked by track changes. Figures 4C, 5C, and S1A are modified, and new Figure S2 is an extension of existing Figure 2E. In addition, molecular weight markers are added to blots in figures 1, 3, 4, and 5.

We have done our best to answer all the points raised by the reviewers. We have however been somewhat restricted in our ability to perform experiments due to the ongoing pandemic.

Reviewer #1 (Evidence, reproducibility and clarity (Required)):

In this manuscript the authors investigate if the localization of Cyclin A2 affects its ability to induce mitosis. This is important since Cyclin A2 cannot induce mitosis during S-phase, where it plays a major role. How Cyclin A2 switches from promoting S-phase to promoting mitosis is not clear. Single cell analysis is being used and complemented by some biochemical experiments. The authors find that cytoplasmic Cyclin A2 can activate PLK1 through the phosphorylation of Bora. Most of the efforts are spent to understand how the subcellular localization of Cyclin A2 is regulated and although the authors are able to exclude a number of possibilities, the final answer is not provided.

This manuscript presents a lot of data and is generally well done. The experiments are described in details, proper controls are included, and most necessary references are listed. I enjoyed reading this manuscript but there are a number of "holes" which keep begging for answers. The authors have done their best to address these questions but have not been able to solve them.

Here are the three most positive points of this manuscript:

1. It addresses an important question in the context of cell cycle
2. The authors used state-of-the-art approaches to investigate this problem
3. The combination of biochemical approaches, single cell analysis, and cellular genetics is powerful

The three most important issues that should be addressed (for details see below):

1. Although it is shown that CycA2/Cdk2 phosphorylates Bora, there is no follow up
2. Inhibitor/siRNA experiments have a number of drawbacks
3. The final answer how the subcellular localization of Cyclin A2 is regulated is missing

There are a number of issues, which need to be addressed:

1. The finding that Bora can be phosphorylated in vitro by CycA2/Cdk2 is interesting but would require more work. For example, which residues are phosphorylated? What is the function of phosphorylation in terms of Bora activity? At this moment, the presented results are interesting but remain in a void that beg to be addressed.

We agree that it would be interesting to perform a full analysis for which residues on Bora are phosphorylated by CDK2 and their importance for PLK1 activation. As CDK1 and CDK2 are similar kinases and both bind to and are directed to targets by the regulatory component Cyclin A2, we suspect that residues phosphorylated by CDK2 will be similar to residues phosphorylated by CDK1, which are well characterized by several studies. We now discuss this possibility in the revised

manuscript (line 192-196). We argue that the manuscript contains a large amount of experiments focusing on the localization change of Cyclin A2, and a thorough analysis of whether there is a difference between Cyclin A2-CDK1 and Cyclin A2-CDK2 mediated phosphorylation of Bora would be better suited for a separate investigation.

2.Despite that the authors work on human cells, they should mention that Cyclin A2 is an essential gene in mice but not in mouse embryonic fibroblasts. Apparently, these cells (and also some human cells) can cycle without Cyclin A2. This is an important background information.

We agree and now include this information (line 68).

3.Line 123: "low EdU staining" (Fig S1C) refers to the wrong figure since there is only DAPI staining in S1C. Maybe this should be Fig 1C?

The graphs to the right in Figure S1C depict a quantification of EdU versus DAPI staining in individual cells.

4.Figure 3 has several issues. The input in (B) is not very helpful since only faint bands are visible in the bottom panel. Probably more protein needs to be loaded there. In addition, there are 2 PLK1 blots but the difference is not explained (in the figure legends). Are these high and low exposures of the same antibody? In (E), a positive control is missing. For example, Cdk2 and/or Cdk1 would work well but others are possible too (p21, p27, etc.). In (F), the authors should include a CycA2 blot. This would be important - maybe the authors already tried this? Finally, just a minor comment to (C), the Autoradiography shows a lot of background, which makes it hard to read since one is distracted by the many bands. Can this be cleaned up?

Figure 3B indeed shows high and low exposures of the same antibody. We now clarify this in the figure legend. The signal is indeed weak, showing that enrichment of PLK1 by IP was successful. This is important as the pT210 antibody can cross-react with other proteins. Although weak, we would argue that the signal is clearly visible in all conditions.

We agree that positive controls are needed and thank the reviewer for finding our mistake of not referring to them. The blots for CDK1, CDK2, and p21 after Cyclin A2 IP are included in Figure 4C instead of Figure 3E. This is to increase readability of the manuscript, as we found that including all in the same blot made the reasoning complex. We now clarify this in the figure legends.

We unfortunately could not blot for Cyclin A2 in figure 3F, as the heavy chain migrates similarly to Cyclin A2.

We agree that there is a lot of signal in the blot. However, as the relevant bands are indicated and most of all complemented by a pT210 staining, we would argue that the results are accessible. Due to the presence of several bands, we prefer to show the entire autoradiography, and not restrict it to certain areas only.

5.Figure 5C: can the amount of Cyclin A2 in the different fractions be quantified? This would be helpful.

We now include a quantification of Cyclin A2 and Cyclin A2-eYFP in figure 5C.

6.Just a comment to the p53 (KO, siRNA) experiments. It is very well known that in the absence of

p53 (but not p21), Cdk expression and activity is massively increased. This should be considered by the authors when they interpret their data.

On line 433-5 we now state that "In addition, absence of negative regulators of CDK activity as p53 and p21 could have secondary consequences due to enhanced CDK activity".

7. Figure S1A: quantification would help.

We have now added quantification to this graph.

8. The experiments with the Cdk inhibitors and the siRNA experiments (for Cdk1 and Cdk2) are interesting but there are many caveats. Therefore, these experiments are far from perfect and the authors should discuss the limitations openly.

We completely agree that that the experiments are interesting but contain many caveats. In addition to our discussion on limitations in the discussion section (for example line 430-433), we now also discuss limitations in the results section on lines 185-188 and 279-282.

9. A number of conclusions are too simplistic or kind of "cherry picking". I would suggest that the authors present several interpretations and they should be careful with their message (since they are often not universally true). I cannot list all of them, but will present a selection. Line 153-154: "This shows that PLK1 activation is impaired after CycA2 depletion, but also suggests that the S/G2 transition is impaired in the absence of CycA2." One alternative interpretation is that PLK1 activation is affected by the S/G2 delay. In other words, this could be an indirect effect and the authors have not proven beyond any doubt that CycA2 depletion is directly regulating PLK1 activation. Line 185: "CycA2-CDK2 can activate PLK1 in vitro and the interactions". I am not sure that the authors have proven this since they only show pT210. This does not seem to be convincing enough. Line 196-201: the interpretation provided is too strong and should be toned down.

There is no intent of cherry picking, and we are sorry to hear that the reviewer perceived the text that way. In fact, line 153-154 was intended exactly as the reviewer points to as an alternative interpretation – that is, as PLK1 is activated during the S/G2 transition and the S/G2 transition is impaired it is not possible to conclude whether it is a direct or indirect effect. Clearly, the phrasing was not optimal, and we now have re-formulated this section. For line 185 and 196-201 we have adapted the reasoning.

10. Line 398: "The levels of CycA2 increase dramatically from early S-phase to late G2-phase." Can the authors provide a reference for this? Even when looking at Figure 1B, I am not sure that I agree with this statement. The peak in mitosis is most likely associated with the rounding up of the cells and does not really reflect an increase in Cyclin A2 molecules. Of course, if you calculate the concentration, this would be technically correct to say there is an increase in mitosis. But does this really matter? In other words, I don't think the CycA2 levels dramatically increase.

Possibly there is a misunderstanding here, as that Cyclin A levels are very low in G1 and then increase until the G2/M transition is well established. In figure 1B, the start of S-phase is not included. Further, we measure average intensity in figure 1B, meaning that the peak in mitosis is affected by cell rounding. In the figure legend we call for caution in interpretation by this reason and we now also include a similar statement in the main text. In contrast, in figure 1C we measure integrated intensity and by EdU staining provide a reference to when S-phase starts. Based on figure 1C Cyclin A2 is hardly detectable when S phase starts, and increases steadily to the end of G2 phase (combining nuclear and cytoplasmic CycA2). We now include a reference to Figure 1C in the statement (now line 437).

Reviewer #1 (Significance (Required)):

This manuscript is good but there are some shortcomings. The authors are quite clear that they have not been able to solve the issue how CycA2 translocates to the cytoplasm. In addition, the Bora aspect has not been completely investigated and would need more work. Therefore, my opinion is that this manuscript is good but of limited significance.

REFEREES CROSS-COMMENTING

I agree with the comments of the other two reviewers

Reviewer #2 (Evidence, reproducibility and clarity (Required)):

****Summary:****

During the cell cycle, CyclinA2 associates with catalytic subunits, Cdk2 and Cdk1, and is required for two key events, DNA replication in S phase and mitotic entry. Previous data support that entry into mitosis is promoted by Aurora-A dependent Polo-like kinase (Plk1) activation that requires the phosphorylation of a co-factor hBora by CyclinA2-Cdk1 and/or -Cdk2. Because hBora is mainly cytoplasmic, how its phosphorylation is timely regulated by CyclinA2-Cdk complexes, which are mainly nuclear in S phase, remains unclear. Here, the authors use K-In U2OS and RPE cell lines with CyclinA2-eYFP fusion and observe CyclinA2 accumulates in the cytoplasm from G2 onset. First, they report that cytoplasmic accumulation is probably not due to changes in chromatin association linked to the termination of the DNA replication process and does not require Cdk2 or Cdk1 activity. Second, they obtained some evidence that cytoplasmic CyclinA2-Cdk2 and/or Cdk1 pool contributes to Plk1 activation. Finally, they found that DNA damage triggers p53 and p21-dependent nuclear CyclinA2 sequestration preceding degradation, preventing entry into mitosis upon genotoxic stress.

****Major comments:****

The cytoplasmic accumulation of CyclinA2 during G2 has been previously described by Zerjatke et al. Cell reports 2017 and must be referenced.

We thank the reviewer for this reference and now cite this paper in the introduction (line 91).

The first conclusion of the present manuscript is that CyclinA2 translocation to the cytoplasm from G2 onset is probably not an active process, but might be due to a gradient effect with the continuous expression/accumulation of CyclinA2 during S-G2 progression. Experiments on the regulation of chromatin association and possible involvement of Cdk2 and Cdk1 activity are convincing and have interest in our understanding of cell cycle regulation. I noticed an unexpected abrupt increase of CyclinA2 expression level in the last hour before mitosis, displayed in Fig1B, which disappears upon Cdk2 or Cdk1 inhibition (sup Fig2a). This might be investigated further and/or discussed by the authors.

In figure 1B, average fluorescence intensity is measured, whereas in figure 1C integrated intensity is measured. As an average measurement is affected by change in cell area, the measurement increases when the cell rounds up and enters mitosis. This increase is not present in figure 1C, in which integrated intensity is measured. In the figure legend we call for caution in interpretation by this reason and we now also include a similar statement in the main text (line 125).

The second conclusion concerns the link between the translocation of CyclinA2 in the cytoplasm from G2 onset and Plk1 activation. A role of CyclinA2 in Plk1 activation, notably through the phosphorylation of Bora, has been recently established by different groups although the importance of the intracellular localization of CyclinA2 was not investigated. To reinforce the claims on this point, it will be important to determine if and how the inducible expression of CyclinA2 specifically in the cytoplasm versus nucleus (Fig2E) affects the activation kinetics of Plk1. Also, the contribution of CyclinA2-Cdk2 versus CyclinA2-Cdk1 to activate Plk1 remains unclear, at least in mammals. The authors report that both Cdk2 and Cdk1 inhibition slow down Plk1 activation in G2 (Fig3A, B), an interesting aspect for cell cycle regulation that will require additional data showing the specificity of the compounds used for Cdk1 versus Cdk2 inhibition in this experimental setting.

We completely agree to these points. First, a direct readout of PLK1 activity after localized Cyclin A-induction would indeed be a powerful experiment. However, this experiment is technically not straightforward and would require a major investment as 1. In the inducible system we rely on expression of GFP and mCherry to identify cells, which precludes the use of a FRET-probe for PLK1 activity. 2. The system requires siRNA-mediated reduction of endogenous Cyclin A2 levels, which impacts on cell cycle progression and precludes the setup described in Figure 1C to estimate kinetics of PLK1 activation. 3. We do not have expression in a large enough proportion of the cells to do a meaningful average readout by Western Blot. Combined with COVID19 restrictions that currently impact on our ability to perform experiments, we therefore chose to on line 417-491 state: *It would be interesting to follow if the spatial and temporal pattern of PLK1 activation differs between expression of wild-type and solely cytoplasmic CycA2.*

Second, CDK inhibitors certainly have limitations, both due to specificity and that CDK2 feeds into production of proteins in, and likely contributes to activation of, feedback loops that stimulate CDK1 activity. On line 185-188, we now state that: *CDK1 and CDK2 are structurally related, and we cannot exclude the possibility that the inhibitor of one CDK in our setup to some extent affects the other CDK. Further, the data does not exclude that the inhibitors target other kinases in addition to CDKs.*

****Minor comments:****

I found the text clear and easy to read. On the other hand, I have recommendations concerning the figures and legends. Marks of molecular weight are missing on all Western blots. Concerning cell fractionation, a marker of the cytoplasmic and nuclear fraction should be displayed (Fig3E, Fig4C). For each figure: 1- it will help to mention in the legend if the experiment was performed with U2OS or RPE CyclinA2-eYFP cell line, 2- if the cells were synchronized or asynchronous (see for example Fig 1E, Fig2E). Concerning IF quantification, how are data normalized? (see Fig2A, B; Fig4B).

We thank the reviewer for spotting that cell lines used were not indicated in all figure legends. These are now updated. We further have updated the Western blots in figure 1, 3, 4, and 5 with molecular weight markers. Figure 3E and 4C (right part) are from the same experiment, and were divided to increase the readability of the manuscript, as we found that including all in the same blot made the reasoning complex. This is now indicated in the figure legends. We now include 14-3-3 and Histone H3 blots to assess fractionation for the left part of Figure 4C.

We also thank the reviewer for spotting that normalization was not described in Figure 2B. We now specify the normalization used in the figure legend. Data in Figure 4B is not normalized between samples.

Reviewer #2 (Significance (Required)):

This work provides valuable information in our understanding of the mechanisms at work that temporally coordinate DNA replication and mitotic entry and will find audience in the cell cycle regulation field. As mentioned above, perturbation assays reinforcing the link between cytoplasmic translocation of CyclinA2 and Plk1 activation will be required.

REFEREES CROSS-COMMENTING

I also agree with the recommendations

Reviewer #3 (Evidence, reproducibility and clarity (Required)):

This manuscript investigates the contribution of a small pool of cytoplasmic Cyclin A2 in regulating progression into mitosis. The authors have established sophisticated live cell imaging models to examine CycA2 localisation in real time and quantitate the changes observed. They provide convincing data for the existence of this small cytoplasmic pool and show that it is decreased in G2 phase checkpoint arrested cells. They also show that the activation of PLK1 in G2 is inhibited by CycA2 depletion, although they fail to provide convincing evidence that this event is controlled by the cytoplasmic CycA2 pool. Beyond this the authors present a lot of interesting experiments but the implications of the data produced are still unclear. It is clear that with depletion of CycA2, over-expression of a constitutively active PLK is insufficient to completely rescue entry into mitosis, although it does increase the rate of entry modestly. Where does the over-expressed PLK1 T210D localise? Is it strictly nuclear? Over-expression of nucleus-directed or cytoplasm-retained CycA2 both partially rescue the G2 phase delay with CycA2 depletion, but is this an indication that both nuclear and cytoplasmic CycA2 is required for full entry into mitosis? The CycA2 6xNES does have a significant nuclear fraction but was little better than CycA2 3xNLS in promoting mitotic entry. So is there a requirement for substantially more nuclear CycA2 for normal mitotic entry? The model present in Figure 7 fails to account for the activator of PLK1, Aurora A. Its localisation is predominantly nuclear in G2 phase, although there must be a small pool of cytoplasmic Aurora A as it localises on the centrosomes in G2 phase. Is the role of cytoplasmic CycA2 to enhance the ability of BORA to collaborate with the small pool of cytoplasmic Aurora A to activate the cytoplasmic pool of PLK1? The in vitro activation data in Figure 3C shows that CycA2-CDK enhances the effects of BORA and Aurora A on PLK1 activation, and this could compensate for the low abundance of Aurora and PLK1 in the cytoplasm. There is a low level of BORA in the nuclear fraction and associated with nuclear PLK1, and even without CycA2 being associated with this complex it is sufficient to strongly activate nuclear PLK1. The PLK1 reporter also shows that PLK1 activation occurs in the nucleus with similar timing or possibly slightly preceding cytoplasmic activation. It has been reported that a role of CycA2 is to co-ordinate mitotic kinase activation in the cytoplasm and nucleus. Is the co-ordination through PLK1 activation at both locations? It is not clear despite the best efforts of the authors to determine what controls the cytoplasmic accumulation of CycA2, although as CycA2 is synthesised in cytoplasm it might be better to ask what regulates the very strong nuclear accumulation of CycA2 in both S and G2 phase.

We agree that there are many issues that remain to be resolved before we understand how CycA2 regulates mitotic entry. We believe that our findings in this manuscript contribute to the general understanding. In this sense, we establish a link between cytoplasmic CycA2 and activation of PLK1, but our data does not support that this link would be the only means by which CycA2 promotes mitotic entry. Similarly, CycA2 is present both in the nucleus and the cytoplasm during G2 phase, and we have no reason to believe that there are not both nuclear and cytoplasmic functions of CycA2.

That said, as the 6xNLS CycA is produced in the cytoplasm, and the 3xNES shows a minor presence in the nucleus (and both versions in principle could shuttle between nucleus and cytoplasm), we would be hesitant to say more than that the experiments point in the direction of what the reviewer suggests.

We agree that the model does not take the localization of Aurora A into account. First, this is as Aurora A can be detected both in the cytoplasm (mainly centrosomes) and in the nucleus. How regulation of Aurora A contributes to mitotic entry and to activation of PLK1 would be an interesting path, but one we believe would be best suited for a follow-up story.

As we discuss in the manuscript, activation of PLK1 exposes an NLS, which relocalises active PLK1 to the nucleus (Kachaner et al., 2017). We therefore are hesitant to draw any conclusions on where PLK1 is activated based on where phosphorylated targets of PLK1 appear. On line 415-417 we now discuss that *It would be interesting to follow if the spatial and temporal pattern of PLK1 activation differs between expression of wild-type and solely cytoplasmic CycA2.*

We agree that it would be interesting to study further the nuclear accumulation of Cyclin A. However, we strongly feel that we have made an extensive exploration into regulation of Cyclin A localization, including aspects of nuclear localization, and believe that addressing additional mechanisms would be better suited for a follow-up project.

The authors have established excellent model systems and shown that a role of CycA2 in regulating S/G2 phase progression is by controlling the activation of G2 phase PLK1. They have also identified a small cytoplasmic pool of CycA2-CDK. However, despite their sterling work they have not managed to demonstrate a role for this cytoplasmic pool, or determined what regulates its appearance. I do not believe that their model is well supported by the data presented.

We thank the reviewer for these words, and would like to emphasize that both in the text and in the graphical model, we specify that additional mechanisms exist.

Specific questions/comments:

What does STLC do? Or etoposide for that matter. It does not appear to affect CycA2 levels at all, at least in Figure S1A?

We now specify the mechanisms for the inhibitors added in the figure legend (line 798-799).

The data in Figure 2B is from 3 cells for each condition. These are difficult experiments but 3 cells is too low a number to make definitive conclusions.

We now clarify that these are representative images based on more cells. We chose to show individual examples rather than quantification of more cells, as we found that it is not straightforward to determine a time-point to use for averaging the data after CycA2 siRNA.

Figure 2E is missing critical non-targeting siRNA controls. Without these it is not possible to determine how well the induced CycA2 proteins rescued the G2 phase delay.

We now include non-targeting siRNA in new Figure S2. Although we agree that it is a useful control, we refrain from specifying how large the rescue is, as CycA2 siRNA delays cells primarily in G2 phase. We feel that a comparison to a population not pre-synchronized by CycA2 depletion would be difficult to quantitatively interpret.

Does p21 inhibiting CycA2-CDK activity destabilise CycA2? If you use CDK1/2 inhibitors on the G2 phase cells do you also see loss of CycA2 in both nuclear and cytoplasm, or only nuclear?

A quantification of nuclear and cytoplasmic CycA2 levels in individual G2 cells after addition of CDK inhibitors is present in Figure S3A. We do not see any major effects on CycA2 levels – at least not within the time-frame of the experiment.

Reviewer #3 (Significance (Required)):

Identified PLK1 as a target for CycA2 in G2 phase, and that CycA2 regulates the S/G2 phase transition. Technically very advanced models and systems.

There are many unresolved issues about how S/G2/M phase transitions are regulated. the role of CycA2 in controlling these transitions is well established, but the mechanism by which this is achieved is still poorly understood. This manuscript provides some important new insights and mechanism.

The work has potentially a broad audience as these kinases are key targets for new anti-cancer drugs.

REFEREES CROSS-COMMENTING

I agree with the comments of the other reviewers

I have worked on Cyclin A-CDK2 and its role in regulating G2 phase progression and G2 phase checkpoint control for over 25 years.

December 15, 2020

RE: Life Science Alliance Manuscript #LSA-2020-00980-T

Dr. Arne Lindqvist
Karolinska Institutet
CMB
von eulers vag 3
P.O. Box 285
Stockholm 171 77
Sweden

Dear Dr. Lindqvist,

Thank you for submitting your revised manuscript entitled "Cyclin A2 localises in the cytoplasm at the S/G2 transition to activate PLK1".

For a brief overview: this manuscript was submitted and reviewed via Review Commons, and the authors transferred their revised manuscript, along with a point-by-point response to the reviewers' concerns to Life Science Alliance.

The revised manuscript was assessed by at least two editors at Life Science Alliance, and was found to sufficiently address the concerns raised. We agree that some of the points raised by the referees would be out of scope.

We invite you to submit a final version of the manuscript for publication in Life Science Alliance, pending minor revisions to comply with the formatting guidelines of Life Science Alliance.

Along with the points listed below, please also attend to the following:

- please upload your main manuscript text as an editable doc file
- please add a Category, Running Title, Summary blurb, and Conflict of Interest statement for your manuscript in our system and in your manuscript text
- please upload your main and supplementary figures as single files
- please upload your Supp. Tables in editable .doc or excel format
- please use the [10 author names, et al.] format in your references (i.e. limit the author names to the first 10
- please add scale bars to Figure 2 A,D, Fig. 4A, Fig. 6A
- please provide the original uncropped gels for the first Kap1 blot in Figure S3C

A. FINAL FILES:

B. MANUSCRIPT ORGANIZATION AND FORMATTING:

Thank you for this interesting contribution, we look forward to publishing your paper in Life Science

Alliance.

Sincerely,

Shachi Bhatt, Ph.D.

Executive Editor

Life Science Alliance

<https://www.lsjournal.org/>

December 23, 2020

RE: Life Science Alliance Manuscript #LSA-2020-00980-TR

Dr. Arne Lindqvist
Karolinska Institutet
CMB
Biomedicum A7
Stockholm 171 77
Sweden

Dear Dr. Lindqvist,

Thank you for submitting your Research Article entitled "Cyclin A2 localises in the cytoplasm at the S/G2 transition to activate PLK1". It is a pleasure to let you know that your manuscript is now accepted for publication in Life Science Alliance. Congratulations on this interesting work.

DISTRIBUTION OF MATERIALS:

Again, congratulations on a very nice paper. I hope you found the review process to be constructive and are pleased with how the manuscript was handled editorially. We look forward to future exciting submissions from your lab.

Happy Holidays and a Happy New Year!!

Sincerely,

Shachi Bhatt, Ph.D.

Executive Editor

Life Science Alliance

<https://www.lsjournal.org/>
